# Disentangling 3D Animal Pose Dynamics with Scrubbed Conditional Latent Variables

**Joshua H Wu**[1,*]  **Hari Koneru**[1]  **J Russell Ravenel**[1]  **Anshuman Sabath**[1]  **James M Roach**[1]
**Shaun SX Lim**[1]  **Michael R Tadross**[1]  **Alex H Williams**[2,3,*]  **Timothy W Dunn**[1*]
[1]Duke University  [2]New York University  [3]Flatiron Institute

## Abstract

Methods for tracking lab animal movements in unconstrained environments have become increasingly common and powerful tools for neuroscience. The prevailing hypothesis is that animal behavior in these environments comprises sequences of discrete stereotyped body movements ("motifs" or "actions"). However, the same action can occur at different speeds or heading directions, and the same action may manifest slightly differently across subjects due to, for example, variation in body size. These and other forms of nuisance variability complicate attempts to quantify animal behavior in terms of discrete action sequences and draw meaningful comparisons across individual subjects. To address this, we present a framework for motion analysis that uses conditional variational autoencoders in conjunction with adversarial learning paradigms to disentangle behavioral factors. We demonstrate the utility of this approach in downstream tasks such as clustering, decodability, and motion synthesis. Further, we apply our technique to improve disease detection in a Parkinsonian mouse model.

## 1 Introduction

Animal behavior is largely expressed through movement (Musall et al., 2019; Bernstein, 1967). Thus, the ability to identify variation in animal movement is critical for understanding neural (dys)function (McCullough & Goodhill, 2021). In recent years, pose estimation has become popular for behavioral measurement (Dunn et al., 2021; Nath et al., 2019; Zimmermann et al., 2020; Bala et al., 2020; Pereira et al., 2022), enabling more objective and comprehensive kinematic profiling. Given the increasing throughput and dimensionality of behavioral measurement assays (Nourizonoz et al., 2020; Bialek, 2022; Marshall et al., 2022), the ability to learn meaningful and interpretable representations of behavior is essential (Datta et al., 2019).

A common analysis goal is to cluster or segment recurring and stereotyped pose sequences—such as running, turning, and rearing—and compare their frequencies and transition statistics across experimental conditions (Berman et al., 2014; Weinreb et al., 2023; Wiltschko et al., 2020; 2015b; Marshall et al., 2022). Currently, most approaches are purely unsupervised and can be sensitive to nuisance variability, such as animal individuality, or over-segment due to continuous variability, such as speed (Van Dam et al., 2023; Costacurta et al., 2022). As a result, these methods often produce action spaces in which true biological signals are uninterpretably entangled with confounding variables or uninformative features. For example, the same fundamental action (e.g., walking) can be performed with different levels of vigor (fast vs. slow) and with different directionality (walking straight vs. slightly turning left or right). All of these variants are typically split into different action clusters (see Marshall et al., 2022), introducing extraneous statistical structure that may obscure more meaningful biological signals.

In principle, action spaces could be made invariant to nuisance variables via disentanglement of specified factors (Shu et al., 2019), thus permitting the identification and analysis of meaningful behavioral dynamics (Shi et al., 2021; Whiteway et al., 2021; Yi & Saxena, 2022). However, existing disentanglement approaches have shortcomings. The increasingly common conditional variational autoencoder (C-VAE) approach learns accurate disentangled representations of the conditional

---

*Correspondence to: {`joshua.wu,timothy.dunn`}@duke.edu, `alex.h.williams@nyu.edu`

variable, such that changing it while holding the other latent variables fixed generates outputs varying only in the conditional factor (Petrovich et al., 2021b; Guo et al., 2020; Khemakhem et al., 2020). However, we will show that the other latent variables are not necessarily, nor typically, invariant to the conditional factor. Adversarial approaches can be used to learn disentangled subspaces with specific invariances (Ding et al., 2020; Sanchez et al., 2020; Zhao et al., 2021; Brakel & Bengio, 2017), but these methods rely on secondary neural networks that can be difficult to train and sensitive to architectural and hyperparameter choices (Arjovsky & Bottou, 2017; Radford, 2015). Furthermore, adversarial neural networks often target fully disentangled representations, e.g., via mutual information minimization (Belghazi et al., 2018; Cheng et al., 2020), but certain nuisance variables, like speed, have *some* dependencies with action understanding (e.g., a subject will always move farther during locomotion than when grooming). Fully disentangling such variables into invariant subspaces would destroy relevant behavioral variation. Ideally, the degree of disentanglement could be easily specified and controlled.

To address these challenges, we present scrubbed C-VAEs (SC-VAE) for learning disentangled motion features from 3D pose sequences (Fig. 1a). Given a variable to disentangle (calculated deterministically or known *a priori*), SC-VAE seeks to learn a latent representation in which the variable cannot be decoded by specified linear or nonlinear functions, called "scrubbers". SC-VAE achieves this by conditioning a VAE on disentanglement variables while adversarially penalizing the encoder based on a scrubber's ability to estimate variable information from the latent representation. As a result, we obtain a variational distribution with direct control over the level of contribution from nuisance variability (Fig. 1b).

We show that scrubbing enhances latent representations by encouraging nuisance-invariant clusters and improving the consistency of the conditional generative model. We also demonstrate SC-VAE's ability to enhance and disentangle biological phenotypes from nuisance factors in a mouse model of Parkinson's disease, illustrating how SC-VAE can be used to further neurobehavioral inquiry. When examining different choices of scrubbing, we find that: 1) neural network scrubbers built from existing adversarial disentanglement methods are less reliable than scrubbers with explicit parametric assumptions, and 2) assumptions of linear or nonlinear disentanglement can impact downstream analyses due to the level of dependence between disentangled variables and behavioral understanding.

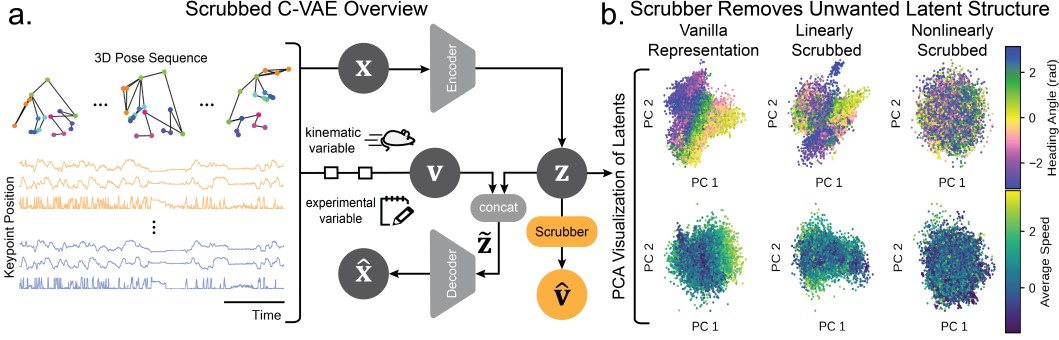

Figure 1: **Overview of SC-VAE.** (a) SC-VAE encourages disentangled representations by directly concatenating known factors to latent variables in a conditional VAE. (b) A "scrubber" removes information about these factors from the latent variables, such that factors are unable to be explained by specified linear or nonlinear functions.

## 2    PRIOR WORK

**Models of animal movement**  Finescale manual annotation (e.g., rears, turns) of animal behaviors is time-intensive and subjective. Further, many experimental manipulations—such as new pharmacological treatments—induce unknown behavioral phenotypes (Nestler & Hyman, 2010). Thus, most research has gravitated towards unsupervised, data-driven methods. Popular approaches apply dimensionality reduction and clustering to wavelet-transformed pose sequences (Berman et al., 2014; Marshall et al., 2021) or auto-regressive hidden Markov models (ARHMMs) to model action motifs as discrete states with intra-state linear dynamics (Wiltschko et al., 2015a; Weinreb et al., 2023).

More recently, deep neural networks have been leveraged to learn low-dimensional representations of behavior. Due to challenging perspective and illumination effects in videos, many behavioral analysis methods use pose trajectories—and increasingly 3D poses (Marshall et al., 2022)—as input (Dunn et al., 2021; Nath et al., 2019; Zimmermann et al., 2020; Bala et al., 2020; Pereira et al., 2022). For example, VAME generates single-frame pose embeddings from variational autoencoders as a nonlinear feature-learning step before fitting an ARHMM (Luxem et al., 2022). BAMS, on the other hand, generates representations by predicting distributions of future kinematics (Azabou et al., 2024).

Our study builds on these works by applying constraints on representational spaces to reduce the contribution of nuisance effects and provide control over these behavioral representations. Our motivation is perhaps most similar to previous work by Costacurta et al. (2022), who developed time-warped extensions to the ARHMM to separate changes in speed from other behavioral factors. Despite similar motivations, our method aims to be a more general tool for disentanglement which learns low-dimensional feature spaces robust to a wider range of factors and that could be used for additional tasks beyond time series clustering.

**Disentangled representations** Although deep networks have the capacity to capture complex nonlinear dynamics, the feature spaces identified by deep networks are not always immediately interpretable and prone to learning undesirable representations due to dataset-specific idiosyncrasies. This unpredictability in learning deep representations has limited their use in the exploratory manner required by experimental research (von Ziegler et al., 2021).

A broad literature on the use of deep networks to extract *disentangled* representations of complex, high-dimensional datasets seeks to address these challenges (Gilpin et al., 2018). While many have investigated the possibility of fully unsupervised disentanglement with VAE-like architectures (e.g., $\beta$-VAE and its variants) (Higgins et al., 2017; Chen et al., 2018; Esmaeili et al., 2019), doing so is challenging in practice. Further assumptions are usually needed to theoretically guarantee identifiability and disentanglement (Locatello et al., 2019), leading to increasing popularity in weakly supervised disentanglement (Shu et al., 2019; Khemakhem et al., 2020).

Neuroscientists have applied and adapted ideas from this literature to account for nuisance variability in raw video data, such as changes in illumination or slight differences in camera angles over days (Whiteway et al., 2021; Shi et al., 2021; Yi & Saxena, 2022). Here we focus on 3D pose data and aim to disentangle additional forms of nuisance variation, such as speed and angular heading, which are cheaply computable from the inferred pose tracks, and animal identity (a type of batch effect). While we focus on these specific variables here, our approach should generalize to other nuisance factors in the future. Prior neuroscience applications of disentanglement have shown success in identifying subspaces for factors. However, they have not explored the invariance over those factors in opposing subspaces.

Our approach is in line with the formalization of disentanglement by Shu et al. (2019) as *both* the ability of: 1) a disentangled dimension to accurately represent a factor, and 2) the corresponding null space to be invariant to that factor. Although there exist disentanglement methods in other application fields that apply both properties (Zhao et al., 2021; Sanchez et al., 2020; Ding et al., 2020), these methods rely on the specification of secondary networks to remove residual factor information in a representation space, often via adversarial training (Ganin & Lempitsky, 2015b; Brakel & Bengio, 2017; Cheng et al., 2020). Here, we contribute novel strategies that: 1) bypass the need for specifying architectures or hyperparameters for adversarial networks, and 2) are more specific about the degree of disentanglement when variables are not perfectly independent of behavioral understanding.

## 3 METHODS

### 3.1 POSE PREPROCESSING

We apply our method to 3D body pose sequences extracted from raw videos. This preprocessing step removes several confounding variables, such as differences in illumination across recording sessions. However, some residual sources of variability remain, such as differences in body size across subjects. Typical methods of pose preprocessing involve translating and rotating all postures to a root position and direction (i.e., "pointing north") (Marshall et al., 2021; Luxem et al., 2022). However, important trajectory and turning information are lost when this alignment is performed on

each frame independently. To avoid this loss of information, we apply a single rotation and translation to all frames in a pose sequence to align the pose at a central time point. That is, given a window of time around a central reference frame, we translate and rotate each posture such that the central frame is at a specified position and orientation (Fig. 4). In a subset of experiments, we refrained from rotating the pose entirely to demonstrate the ability of our method to disentangle the heading direction and recapitulate a representation commensurate with the fully processed data.

In contrast to some existing methods (e.g., Luxem et al., 2022), we represent pose using joint angles between keypoints positioned at fixed distances from their neighbors. Intuitively, this representation forces any pose dynamics model to obey the physical constraints of the body skeleton, leading to cleaner generative samples (see discussion in Marshall et al., 2021; Huang et al., 2017; Vemulapalli et al., 2014). Optimizing over raw angular variables is complicated by discontinuities; thus, we follow a well-established protocol from the computer vision community for lifting 3D rotation matrices into a continuous 6D space (Zhou et al., 2019). Each posture composed of $K$ joint angles is then represented as a $(6K + 3)$-dimensional vector, where the three additional dimensions come from the global position of the root keypoint. Altogether, each pose sequence is denoted $\mathbf{x}_t \in \mathbb{R}^{L \times (6K+3)}$ for a sequence over $L$ frames centered at frame $t$.

## 3.2 DISENTANGLED REPRESENTATIONS AND CONDITIONAL VAE ARCHITECTURE

**Disentangled variables** Our core goal is to learn a low-dimensional representation of behavior where simple factors (e.g., speed, heading direction, and animal identity) are put into orthogonal dimensions from factors that are learned in a data-driven, unsupervised fashion. In essence, this would separate the semantic content of actions—whether the animal is running, digging, rearing, etc.—from the character of these actions—whether the animal is running fast vs. slow or oriented differently when turning.

We formalize this goal as follows. We assume that for any pose sequence $\mathbf{x}_t$ we can compute a vector of nuisance factors $\mathbf{v}_t \in \mathbb{R}^N$. Loosely, we will say that a latent representation $\tilde{\mathbf{z}}_t$ taking values in $\mathbb{R}^D$ is *disentangled* with respect to $\mathbf{v}_t$ when one can identify two orthogonal subspaces $\mathbf{z}_t$ and $\mathbf{z}'_t$ such that $\mathbf{v}_t$ can be reliably inferred from $\mathbf{z}'_t$ but not reliably inferred from $\mathbf{z}_t$. In other words, the goal is to have $\mathbf{z}'_t$ capture known behavioral factors, while $\mathbf{z}_t$ reflects unknown behavioral factors that are learned from the data with minimal redundancies to $\mathbf{z}'_t$.

**Variational autoencoders (VAEs)** The backbone of our method is a VAE with 1D convolutional layers (Kingma & Welling, 2013). Adopting standard notation, we use an encoder network $q_\phi$ to map pose sequences onto a distribution over latent variables sampled via a reparameterization trick, $\mathbf{z}_t \sim q_\phi(\cdot \mid \mathbf{x}_t)$. The pose sequence is then reconstructed by a decoding network which evaluates the likelihood $p_\theta(\mathbf{x} \mid \mathbf{z}_t)$. In a standard VAE, the parameters $\{\phi, \theta\}$ are adjusted with respect to the evidence lower bound (ELBO):

$$L_{\text{ELBO}}(\phi, \theta) = -\mathbb{E}_{\mathbf{x}_t}\left[\mathbb{E}_{\mathbf{z}_t \sim q_\phi(\cdot|\mathbf{x}_t)}\left[\log p_\theta(\mathbf{x}_t, \mathbf{z}_t) - \log q_\phi(\mathbf{z}_t \mid \mathbf{x}_t)\right]\right] \quad (1)$$

**Conditional VAEs (C-VAEs)** An established way to promote disentanglement in a supervised fashion is to use a conditional VAE (Sohn et al., 2015; Khemakhem et al., 2020). C-VAEs are a popular approach for controllable human motion synthesis, as they can generate realistic body pose sequences that correspond to categorical actions (e.g., throwing, kicking, or lunging, Guo et al., 2020; Petrovich et al., 2021a; Gu et al., 2024). C-VAEs have also been exploited within neuroscience (Wu et al., 2020), but to our knowledge, have not been leveraged for behavioral representation learning.

Here, latent variables are sampled from a conditional prior distribution that depends on another variable $\mathbf{v}_t$, which is observed separately or derived deterministically from $\mathbf{x}_t$. In our case, $\mathbf{v}_t$ will correspond to a vector representation of the animal's running speed, heading direction, or one hot encoding of the animal's identity. Exact implementations for C-VAEs vary. We choose a simple approach of concatenating $\mathbf{v}_t$ onto the sampled output of the encoder network. That is, we sample $\mathbf{z}_t \sim q_\phi(\cdot \mid \mathbf{x}_t)$ while $\mathbf{z}'_t$ is strictly defined by $\mathbf{v}_t$. A new vector $\tilde{\mathbf{z}}_t = [\mathbf{z}_t \quad \mathbf{v}_t]^\top$ is passed to the decoding network. The C-VAE is then trained with respect to the same ELBO loss, which is now interpreted as a lower bound of the expected conditional marginal likelihood, conditioning with respect to $\mathbf{v}_t$. For full details on model architectures, see C.1.

Intuitively, since $\mathbf{z}_t$ and $\mathbf{v}_t$ are both passed to the decoder, any information about $\mathbf{v}_t$ that is encoded by $\mathbf{z}_t$ is redundant and inefficient. Thus, if the latent dimension $D$ is small, one expects that the C-VAE

will learn a disentangled representation. While this intuition is borne out to a certain extent in practice, our experience (documented below in Sec. 4) shows that a C-VAE will often learn a distribution over $\mathbf{z}_t$ that is only partly disentangled from $\mathbf{v}_t$. Further, choosing a smaller latent dimension $D$ can be expected to result in more efficient coding and disentanglement but less expressivity and trainability. Thus, a vanilla C-VAE seems to impose a trade-off between faithful modeling of the data and learning disentangled latents.

### 3.3 Scrubbing out residual information

A core component of our approach is to augment the C-VAE loss function to reduce dependence between $\mathbf{z}_t$ and $\mathbf{v}_t$. This is analogous to minimizing mutual information (MI), $I(\mathbf{z}_t, \mathbf{v}_t) = \mathbb{E}_{\mathbf{z}_t, \mathbf{v}_t} \left[ \log(p(\mathbf{z}_t, \mathbf{v}_t)/p(\mathbf{z}_t)p(\mathbf{v}_t)) \right]$. Although MI maximization is commonly used to generate representations with shared features between paired samples (Hjelm et al., 2018; Bachman et al., 2019; Zhao et al., 2021; Sanchez et al., 2020), there are generally no tractable upper bounds for minimization (Poole et al., 2019). In the context of behavioral analysis, our objective is to create a representation space in which $\mathbf{v}_t$ is "hidden" from downstream analyses with constrained expressiveness (e.g., linear regression or clustering via a Gaussian mixture model). We therefore substitute MI with predictive information (Xu et al., 2020b):

$$I_f(\mathbf{z}_t, \mathbf{v}_t) = \max_\psi \left[ \mathbb{E}_{\mathbf{x}_t, \mathbf{v}_t} \left[ \mathbb{E}_{\mathbf{z_t} \sim q_\phi(\cdot | \mathbf{x}_t)} \left[ \log p(\mathbf{v}_t \mid f_\psi(\mathbf{z}_t)) \right] \right] \right] \tag{2}$$

where $f_\psi(\cdot)$ is a potentially nonlinear function family parameterized by $\psi$, and $\log p(\mathbf{v}_t \mid f_\psi(\mathbf{z}_t))$ is the log-likelihood of $\mathbf{v}_t$ conditioned on sufficient statistics $f_\psi(\mathbf{z}_t)$. Given an approximate maximizer $\hat{\psi}$ of eq. (2), we promote disentanglement by introducing an adversarial penalty $L_{\text{Scrub}}(\phi) = \hat{I}_f(\mathbf{z}_t, \mathbf{v}_t)$. We then seek to jointly minimize $L_{\text{ELBO}}(\phi, \theta) + \lambda L_{\text{Scrub}}(\phi)$ over the model parameters $\{\phi, \theta\}$ for some user-specified hyperparameter $\lambda > 0$.

Thus, $\hat{f}_\psi(\cdot)$ is an adversarial decoder that aims to maximize the log-likelihood of $\mathbf{v}_t$ given $\mathbf{z}_t$. This penalty term is minimized when the encoder network, parameterized by $\phi$, encodes a variational distribution over $\mathbf{z}_t$ with no decodable information about $\mathbf{v}_t$. This would imply perfect disentanglement with respect to the class of functions parameterized by $\psi$. Mutual information is recovered in the limit that $f_\psi(\cdot)$ defines an arbitrarily complex probability density (see Xu et al., 2020b), but we will see that it can be advantageous to constrain $f_\psi(\cdot)$ to a more limited class of functions (e.g., linear).

The effect of the penalty term $\lambda L_{\text{Scrub}}(\phi)$ is to "scrub out" information about $\mathbf{v}_t$ contained in $\mathbf{z}_t$, and we therefore call $f_\psi(\cdot)$ a "scrubber." The overall model architecture is diagrammed in Fig. 1a. However, it is challenging to efficiently solve the inner maximization problem over $\psi$ in eq. (2). We now describe several strategies to overcome this challenge.

### 3.4 Scrubbing with adversarial neural networks

A classic approach to adversarial training is to define $f_\psi$ to be a neural network trained jointly with the C-VAE. We implement two such methods here for disentangling behavioral representations.

**Scrubbing with gradient reversal layers (SC-VAE-GR)** First, we apply an existing method that is popular in the context of domain adaptation in deep networks (Ganin & Lempitsky, 2015a) and applied to disentanglement (Ding et al., 2020). Where $f_\psi(\cdot)$ is a multi-layer perception (MLP) ensemble, the idea is to estimate the gradient of:

$$\mathbb{E}_{\mathbf{x}_t, \mathbf{v}_t} \left[ \mathbb{E}_{\mathbf{z_t} \sim q_\phi(\cdot | \mathbf{x}_t)} \left[ \log p(\mathbf{v}_t \mid f_\psi(\mathbf{z}_t)) \right] \right] \tag{3}$$

with respect to $\psi$ and $\phi$ over a minibatch of pose sequences. Then, we simultaneously perform gradient ascent and descent updates on $\psi$ and $\phi$, respectively. This is akin to jointly training the model over $\{\psi, \phi, \theta\}$ but with the gradients reversed for $\psi$.

**Scrubbing with neural discriminators (SC-VAE-ND)** An alternative approach investigated by Sanchez et al. (2020), Brakel & Bengio (2017), and Kim & Mnih (2018) draws inspiration from generative adversarial networks (GANs; Goodfellow et al., 2020). Given a latent sample $\begin{bmatrix} \mathbf{z}_i & \mathbf{v}_i \end{bmatrix}^\top$, we generate a "fake" sample $\begin{bmatrix} \mathbf{z}_i & \mathbf{v}_{j \neq i} \end{bmatrix}^\top$ and define an adversarial discriminator, $f_\psi(\cdot)$, to classify real from fake representations. By minimizing the cross-entropy loss of this classifier, the mutual

information between $\mathbf{z}_t$ and $\mathbf{v}_t$ is minimized (Brakel & Bengio, 2017). Unlike for the rest of the methods in this paper, $\hat{f}_\psi(\cdot)$ and the C-VAE are trained in alternation for SC-VAE-ND.

**Challenges** These strategies can work if one carefully tunes separate learning rates for $\psi$ and $\{\phi, \theta\}$. However, we found that fine-tuning these hyperparameters was, perhaps unsurprisingly, challenging in practice (Shen et al., 2020; Scaramuzzino et al., 2023). When not delicately tuned, $\hat{\psi}$ fails to track the distribution of $\mathbf{z}_t$ induced by parameter updates to the encoder $q_\phi$. Thus, $\hat{f}_\psi(\cdot)$ will be a suboptimal adversary, and $\hat{I}_f(\cdot)$ will not be tight to $I_f(\cdot)$. Furthermore, these approaches have strong assumptions of nonlinear disentanglement, which may not be appropriate in cases where the scrubbed variable is not independent (see Sec. 4.2 and 4.4). In the following methods, we navigate these challenges by: 1) adaptively tuning learning rates, 2) applying more direct estimations of $\psi$ or $I_f(\cdot)$ which do not require specifying network architectures and their hyperparameters, and 3) introducing scrubbers with more constrained (e.g., linear or quadratic) assumptions of scrubbing.

### 3.5 SCRUBBING UNDER PARAMETRIC CONSTRAINTS

**Linear scrubbing by moving average least squares (SC-VAE-MALS)** If we constrain $f_\psi(\mathbf{z}_t)$ to be a linear function and use a Gaussian likelihood to model $\mathbf{v}_t$, then the maximization over $\psi$ is simply an ordinary least squares problem. The optimal prediction can be calculated in closed-form by the normal equations (see e.g., Murphy, 2012, Ch. 7):

$$f_\psi(\mathbf{z}_t) = \mathbb{E}[\mathbf{v}\mathbf{z}^\top]\left(\mathbb{E}[\mathbf{z}\mathbf{z}^\top] + \beta\mathbf{I}\right)^{-1}\mathbf{z}_t \tag{4}$$

where $\beta$ is the L2-regularization constant. Thus, to evaluate $L_{\text{Scrub}}(\phi)$ for a linear model, we simply need to keep a running estimate of two matrices: $\psi^0 = \mathbb{E}[\mathbf{v}\mathbf{z}^\top]$ and $\psi^1 = \mathbb{E}[\mathbf{z}\mathbf{z}^\top]$. We do this by computing exponential moving averages (EMA) of these quantities, replacing expectations with empirical averages. Given a single data instance $\mathbf{x}_t$, we sample $\mathbf{v}_t, \mathbf{z}_t \sim q_\phi(\cdot \mid \mathbf{x}_t)$, and update our estimate of the sufficient statistics accordingly:

$$\hat{\psi}^0 \leftarrow [(1-\alpha)\mathbf{v}_t\mathbf{z}_t^\top + \alpha\hat{\psi}^0] \tag{5}$$

$$\hat{\psi}^1 \leftarrow [(1-\alpha)\mathbf{z}_t\mathbf{z}_t^\top + \alpha\hat{\psi}^1] \tag{6}$$

where $\alpha$ is the smoothing factor of this EMA filter. In practice, we compute these updates over a minibatch with multiple data samples. We forgo manual tuning of $\alpha$ by running two EMA filters in parallel with separate smoothing factors at a small, fixed difference. On each minibatch, we increment or decrement the smoothing factors based on which filter provides a better fit to the minibatch statistics (see A.2 for details).

**Extension to scrubbing by polynomial regression** The method above immediately extends to polynomial regression by augmenting $\mathbf{z}_t$ to include higher-order polynomial features to obtain quadratic (-MAQS) and cubic (-MACS) scrubbing. This enables progressive nonlinear scrubbing.

**Categorical scrubbing with adaptive quadratic discriminators (SC-VAE-QD)** The EMA filter can also be readily applied to other scrubbing models beyond ordinary least squares. When $\mathbf{v}_t$ is a categorical variable encoding animal identity, we treat $f_\psi(\mathbf{z}_t)$ as a prediction of the class label. A simple approach is to keep a running estimate of the class-conditional means, $\mathbb{E}_{\mathbf{v}=c}[\mathbf{z}]$, and covariances, $\text{Cov}_{\mathbf{v}=c}[\mathbf{z}, \mathbf{z}]$, and then use a quadratic discriminant analysis (QDA) classifier $\hat{f}_\psi(\mathbf{z}) = \hat{p}(\mathbf{v}_t = c|\mathbf{z}_t)$ (see A.2). We update the mean and covariance estimates in the same way as eq. (5) and eq. (6) with the self-tuning smoothing factor.

**Nonlinear scrubbing with kernel density mutual information estimators (SC-VAE-MI)** The constrained adversarial maximization over $f_\psi(\cdot)$ is related to classical quantities in information theory. In particular, if the class of functions parameterized by $\psi$ contains all possible models, then minimizing $L_{\text{Scrub}}(\phi)$ is equivalent to minimizing the mutual information between $\mathbf{z}_t$ and $\mathbf{v}_t$ (Xu et al., 2020a). Above, we built scrubbers assuming a parametric form for $f_\psi(\cdot)$ (e.g., linear or quadratic), but we can also develop a scrubber that estimates the mutual information in a nonparametric fashion. We investigated a simple approach of using Gaussian kernel density estimation to approximate the joint probability density $\hat{p}(\mathbf{z}_t, \mathbf{v}_t)$. Given this approximation of the density, $\hat{I}(\mathbf{z}_t, \mathbf{v}_t) = \mathbb{E}_{\mathbf{z}_t, \mathbf{v}_t}[\log(\hat{p}(\mathbf{z}_t, \mathbf{v}_t)/\hat{p}(\mathbf{z}_t)\hat{p}(\mathbf{v}_t))]$ provides an estimate of the mutual information (Moon et al., 1995). It is well-known that mutual information estimators suffer a curse of dimensionality;

to combat this, we use a large batch size (2048 pose sequences) and tune the bandwidth parameter of the Gaussian kernel to compromise between estimator bias and variance. This results in a biased estimate of mutual information, which we identify with $L_{\text{Scrub}}(\phi)$.

# 4 RESULTS

## 4.1 DATA

We used data from n = 4 C57BL/6 mice freely exploring a 30 cm$^2$ open field (324k frames per mouse, 1 hour at 90 fps). For applications, we used n = 36 C57BL/6 mice recorded in 1-hour sessions (90 fps) in this same open field before and after Parkinson's disease (PD) induction (6-OHDA unilateral injection model). 3D poses were tracked with DANNCE (Dunn et al., 2021). In each animal, we also performed fluorescent immunostaining of the striatum bilaterally and quantified the extent of dopaminergic neurodegeneration as the ratio of integrated fluorescence between the lesioned to non-lesioned hemispheres (more details in D).

## 4.2 DISENTANGLING NUISANCE VARIABLES

For all results, we trained separate models to isolate the effects of each scrubber on each nuisance variable. For implementation details, see B and C.

**Heading direction** We first tested heading direction disentanglement in models trained *without* rotationally processed data as described in Section 3.1. As the optimal representation can be derived by fitting a VAE to fully processed pose sequences (VAE Processed), heading disentanglement serves as an ideal baseline to validate our scrubbing methods. When examining a vanilla VAE trained on raw (not rotationally-processed) data, heading direction was, unsurprisingly, linearly decodable from the latents (Fig. 2a). Conditioning the latent variable was not sufficient to disentangle heading direction from other latent factors; a C-VAE conditioned on heading direction formed representations that were structured with heading direction in a similar way to the vanilla VAE, despite being redundant with the conditioned variable. Although SC-VAE-GR and -ND use nonlinear adversaries (i.e., an MLP), we found they were unreliable in producing representations that were even linearly invariant to heading. SC-VAE-MALS was found to be more reliable in achieving linear disentanglement by directly estimating a linear adversary.

Although SC-VAE-MALS was successful in linear disentanglement, an optimal representation arguably should have near complete disentanglement (beyond linear) of heading direction since the facing direction of a behavior does not affect its semantic label.[1] Scrubbing by quadratic regression (SC-VAE-MAQS) or mutual information estimation (SC-VAE-MI) reached nonlinear decodability commensurate with the rotationally preprocessed optimum (VAE Processed). Thus, both SC-VAE-MI and -MAQS exhibit strong linear and nonlinear disentanglement. We then tested SC-VAE disentanglement on speed and animal identity: nuisance variables whose influences on behavioral representations cannot be trivially removed via data preprocessing.

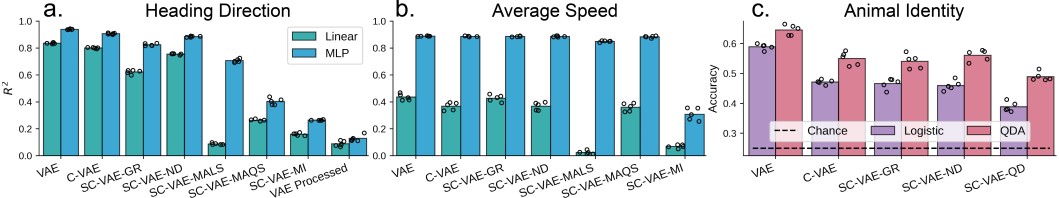

Figure 2: **Decoding nuisance variables from scrubbed latents.** $R^2$ for linear and nonlinear (MLP) regression on (a) speed and (b) heading from latents. (c) Classification accuracy for animal identity using logistic regression and quadratic discriminants.

---

[1]Importantly, we will argue below that this logic does not apply to some other behavioral variables, such as the animal's speed, which are *not* independent of action categories. In these cases, fully nonlinear scrubbing may lead to undesirable performance, but a constrained (e.g., linear) scrubber can provide partial disentanglement.

**Speed**  Unlike with heading direction, speed is naturally entangled with behavioral factors—e.g., rearing and running exhibit inherently different speed profiles. The generative model defined by the VAE ought to reflect this dependence, so *some* nonlinear dependence between speed and the unsupervised latents should be preserved and not "scrubbed out". On the other hand, the same action can be performed at different speeds, so it would be desirable to at least partially disentangle speed from other variables. Trends for linear and nonlinear disentanglement across models for speed scrubbing were similar to our heading results (Fig. 2b). SC-VAE-MAQS was the exception where it did not affect speed disentanglement. Although SC-VAE-MI continued to exhibit strong disentanglement, this came at the expense of having interpretable clusters of actions within the latent space. We further explore the subtleties of linear vs. nonlinear speed disentanglement in Section 4.4.

**Animal identity**  Small differences in subject limb proportions or sizes can affect behavioral kinematics and postures in complex nonlinear ways, even in pose preprocessing without explicit notions of limb lengths (e.g., ours described in Sec. 3.1). These differences cause trivial separations of the same behaviors across individuals. When scrubbing animal identity, SC-VAE representations were also the most linearly and nonlinearly invariant (Fig. 2c). As with heading direction and speed, neural network scrubbers (SC-VAE-GR and -ND) were less effective than using a specified nonlinear scrubber (SC-VAE-QD) for identity disentanglement.

Scrubbing effects were evident when visualizing latent representations (Fig. 1b, 6). Vanilla VAE and C-VAE representations were highly entangled with nuisance variables across principal axes, with regions dominated by specific speeds, heading directions, or animal identities. The scrubbing models showed representations with more mixing across nuisance variables. SC-VAE-MI, for heading and speed, and SC-VAE-QD, for identity, produced the most "salt and pepper" representations.

Table 1: **Effects of scrubbing on motion synthesis.** $R^2$ of heading or speed of generated behaviors with respect to random heading or speed conditioned inputs to the decoder.

| Model | Heading Direction $R^2$ ↑ | Average Speed $R^2$ ↑ |
|---|---|---|
| C-VAE | 0.502 | 0.635 |
| SC-VAE-GR | 0.957 | 0.654 |
| SC-VAE-ND | 0.079 | 0.626 |
| SC-VAE-MALS | 0.983 | 0.683 |
| SC-VAE-MAQS | 0.989 | 0.648 |
| SC-VAE-MI | 0.994 | 0.607 |

### 4.3  CONSISTENT MOTION SYNTHESIS

To evaluate whether our models in Section 4.2 maintained valid representations when scrubbing heading or speed decodability, we conditioned their decoders on random speeds and heading directions and evaluated the resulting speeds and headings of the generated sequences (Table 1). We found that scrubbing improved the consistency of conditionally generated sequences, suggesting new strategies for improving conditional motion synthesis.

### 4.4  CLUSTERING DISENTANGLED REPRESENTATIONS

When clustering with Gaussian mixture models (GMMs), we found that the anisotropies due to nuisance variability in vanilla VAE and C-VAE representations determined the makeup of behavioral clusters, as they were comprised largely of behaviors with similar speeds or heading directions (Fig. 3a, e) as opposed to typical behaviorally relevant clusters (Fig. 7). Thus, these nuisance variables drive a trivial segmentation of behavioral space that obscures biologically meaningful behavioral identification. Scrubbing the C-VAE representations progressively reduced the influence of nuisance variables, increasing the variance of heading direction and speed within GMM clusters (Fig. 3b, f *top*). In the heading direction test case, SC-VAE-MI completely eliminated nuisance clustering and produced fully heading-mixed behavioral clusters. (Fig. 3d).

However, we found that MI scrubbing for speed was too strong and produced clusters no longer associated with typical coarse behavioral labels (e.g., walking, rearing, grooming). To further investigate, we trained a second vanilla VAE and labeled six walking clusters with differing distributions of speed (Fig. 3e). When comparing the latents in other models associated with these labeled walking clusters, the distributions of walking behaviors in the SC-VAE-MALS and SC-VAE-GR representa-

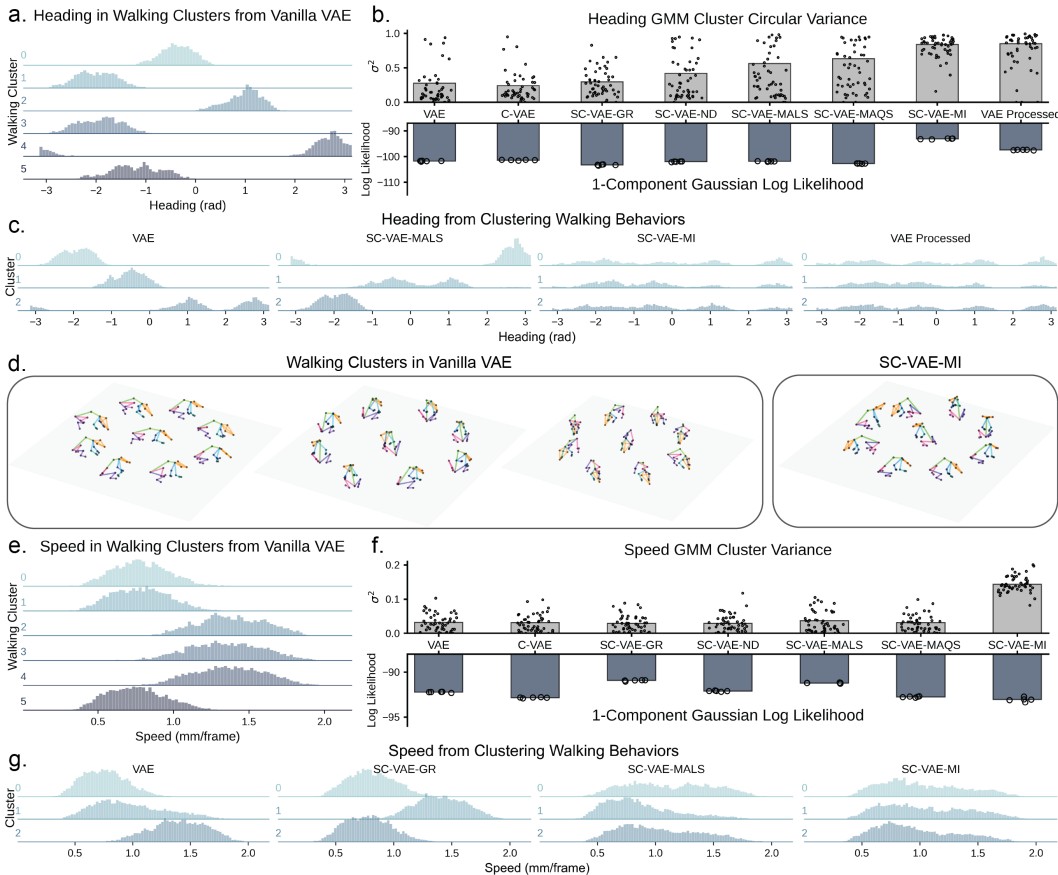

Figure 3: **Examining the effects of scrubbing on clustering.** (a) Heading direction distribution in walking GMM ($k = 50$) clusters identified from a separate VAE trained on not rotationally preprocessed data. (b) *(top)* Weighted average circular variance of heading by cluster. *(bottom)* 5-fold cross log-likelihood of a diagonal Gaussian component on the normalized latents of each model corresponding to the clusters in (a). (c) Histograms of heading in GMM clusters of VAE, SC-VAE-MALS, -MI, and VAE Processed latents corresponding to the clusters in (a). (d) Example walking clusters in VAE *(left)* and SC-VAE-MI *(right)* after scrubbing heading. (e) Speed distribution in walking GMM ($k = 50$) clusters identified from a separate VAE. (f) *(top)* Weighted average variance of speed by cluster. *(bottom)* 5-fold cross log-likelihood of a diagonal Gaussian component on the normalized latents per model corresponding to the clusters in (e). (g) Histograms of average speed in GMM ($k = 3$) clusters of VAE, SC-VAE-GR, -MALS, and -MI latents corresponding to the clusters in (e).

tions were better captured by a single cluster than other models (Fig. 3f *bottom*), demonstrating how scrubbing can combine behaviors previously distinguished only by nuisance variation (see B.6 for details on this metric). Further clustering of these walking behaviors in SC-VAE-MALS revealed completely speed-mixed clusters, whereas the separation was still clearly present in vanilla VAE. Repeating this with heading scrubbing (separately identifying six walking clusters with different heading distributions), SC-VAE-MI was found to have the most ideal walking representation (Fig. 3c, d). These results suggest the need for disentanglement methods with weaker (-MALS) and stronger (-MI) assumptions for disentanglement which reflect the level of independence between nuisance variables and behavioral understanding.

## 4.5 NEUROLOGICAL DISEASE ANALYSIS

Subject-specific idiosyncrasies increase variability in behavioral expression across animals, potentially obscuring the effects of neurobehavioral perturbations. In our PD dataset, we found that scrubbing

animal identity attenuated this variability, enhancing the detection of behavioral shifts in disease (PD) sessions. Using maximum mean discrepancy (MMD; B.6; Gretton et al., 2012; Goffinet et al., 2021) to measure behavioral distribution similarity between recording sessions, we observed a large increase in similarity within conditions and a decrease between conditions for the SC-VAE-QD model (Fig. 9). That is, a given disease session moved closer to the other disease sessions and farther from the healthy sessions, suggesting improved discriminability of disease-associated behavioral changes. We quantified this using an effect size metric, $d$, calculated as the MMD between a subject's healthy and disease sessions relative to the MMD between healthy sessions across animals (Table 2 *left*, B.6). We also used a k-NN classifier to predict the type of session (healthy vs. disease) given a behavioral representation (Table 2 *middle*). SC-VAE-QD outperformed other models on these metrics. As a control, we also scrubbed the healthy vs. disease session label itself ('Reverse Control'), which resulted in a significant decrease in all disease discriminability metrics.

Striatal immunostaining revealed another potential source of PD phenotype variability: not all animals experienced the same degree of dopaminergic denervation. More intact dopaminergic circuity should be associated with fewer behavioral changes between healthy and PD sessions (Boix et al., 2015; Slézia et al., 2023). Thus, we expect behavioral representations better capturing these changes to be more correlated with the extent of denervation. While the healthy-disease MMD was correlated for all models, SC-VAE-QD strengthened the correlation (Table 2 *right*).

Table 2: **Scrubbing identity in a Parkinsonian mouse dataset.** Impact of identity scrubbing quantified as effect size ($d$) mean $\pm$ SEM across animals, healthy vs. disease classification accuracy (%), and the correlation (Pearson $r$) of MMD behavioral change to dopamine denervation extent.

| Model | Effect Size $d \uparrow$ | Accuracy (%) $\uparrow$ | Denervation correlation ($r$) $\uparrow$ |
|---|---|---|---|
| VAE | $1.07 \pm 0.05$ | 81.3 | 0.208 |
| C-VAE | $1.11 \pm 0.05$ | 82.2 | 0.221 |
| SC-VAE-GR | $1.17 \pm 0.06$ | 83.3 | 0.249 |
| SC-VAE-QD | $1.71 \pm 0.11$ | 85.8 | 0.406 |
| Reverse Control | $0.68 \pm 0.03$ | 65.2 | 0.086 |

## 5 CONCLUSION AND LIMITATIONS

This study presents a novel framework for disentangling nuisance behavioral or experimental variables by removing (i.e., "scrubbing") variable information from latent spaces via adversarial learning objectives. Our adaptive moving average, quadratic, and mutual information scrubbers are successful in producing behavioral representations that are linearly or nonlinearly invariant to nuisance variables, reducing behavioral over-segmentation and enhancing the interpretability of latent spaces. Existing approaches based on gradient reversal and adversarial networks generally only assume nonlinear scrubbing and, in our experience, are difficult to fine-tune. In contrast, our scrubbing approach using exponential moving averages of sufficient statistics is hyperparameter-free.

A limitation of our convolutional encoder-decoder architecture is that it represents fixed-length pose sequences as a single latent code, which constrains feature timescales and does not scale efficiently to long sequences. This issue could be addressed in future work by adopting a dynamic latent variable model. Additionally, our nuisance variable scope can be expanded to include factors on finer spatial and temporal scales, such as per frame or individual body part kinematics. Although all of the models presented in this paper were trained while scrubbing one nuisance variable at a time, SC-VAE can, in principle, be trained to jointly condition on multiple nuisance variables. This could increase the purity of behavioral representations, enable multifactor motion synthesis, and permit direct comparisons of nuisance variable effects within a common space. Future work should explore the effects of scrubbing multiple nuisance variables, either jointly or individually.

Finally, we have focused on scrubbing out nuisance variables that are easily available (e.g., speed, heading direction, and animal identity), but it may not be obvious how to identify analogous variables or determine the correct level of disentanglement in more complex datasets (e.g., social). Future work could focus on scrubbing in exploratory analyses, such as to fine-tune fully-unsupervised models. This work and future work seek to address the growing necessity for predictable and interpretable deep representation learning methods for neuroscientific research.

## 6 ACKNOWLEDGEMENTS

J.H.W. acknowledges support from the National Science Foundation Graduate Research Fellowship under Grant No. DGE 2139754 and the supplemental INTERN Grant No. DGS 2139754 Amendment 007. M.R.T. acknowledges support from the National Institutes of Health (NIH) grant R34DA059512, and T.W.D. from NIH grants R34DA059512 and R01GM136972.

## 7 ETHICS STATEMENT

*Animal care and use:* The care, procedures, and experimental manipulation of all animals were reviewed and approved by Duke University Institutional Animal Care and Use Committee. See D for further details. *Declaration of interests statement:* T.W.D. is a co-founder of dannce.ai.

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

## A    METHOD DETAILS

### A.1    POSE PREPROCESSING

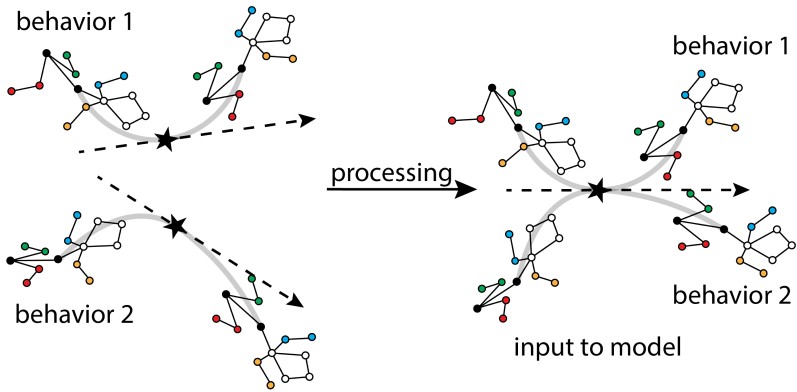

Figure 4: **Aligning pose trajectories.** All pose sequences are translated and aligned so that the central frame is globally centered and rotated to face in a standard direction. This representation preserves important trajectory information. We decompose our pose sequences into local joint rotations and global root position as is commonly done in 3D deep pose methods (e.g., Petrovich et al., 2021b; He et al., 2022).

### A.2    SCRUBBERS

**Scrubbing by adaptive moving average least square (SC-VAE-MALS)**

Consider a linear decoder:

$$f_\psi(\mathbf{z}) = \psi \mathbf{z} = \hat{\mathbf{v}} \tag{7}$$

where $\psi = \psi^0(\psi^1 + \beta \mathbf{I})^{-1}$, which can be evaluated using the loss:

$$L(\mathbf{z}, \mathbf{v}; \psi) = ||\mathbf{v} - f_{\psi_a}(\mathbf{z})||_2^2 \tag{8}$$

---

**Algorithm 1** SC-VAE-MALS

---

Initialize network parameters
$\phi, \theta$
Initialize linear estimator parameters
$\hat{\psi} \in \mathbb{R}^{N \times D}$
Initialize forgetting factor
$\alpha \in (0, 1)$
Initialize L2-regularization coefficient
$\beta \in \mathbb{R}$
**repeat**
  Draw batch with $K$ samples
  $\{\mathbf{x_k}, \ \mathbf{v_k} \in \mathbb{R}^N, \ \mathbf{z_k} \sim q_\phi(\cdot \mid \mathbf{x_k}) \in \mathbb{R}^D\}_{\mathbf{k}=1}^K$
  Calculate decoder loss:
  $L_{\text{Scrub}} = \frac{1}{K} \sum_{\mathbf{k}=1}^K L(\mathbf{z_k}, \mathbf{v_k}; \psi)$
  Update $\psi$ based on the normal equations for ordinary least squares regression:
  $\hat{\psi}_{k+1}^0 = (1-\alpha) \sum_{\mathbf{k}=1}^K (\mathbf{v_k}\mathbf{z_k}^\top) + \alpha \psi_k^0$
  $\hat{\psi}_{k+1}^1 = (1-\alpha) \sum_{\mathbf{k}=1}^K (\mathbf{z_k}\mathbf{z_k}^\top) + \alpha \psi_k^1$
  Update network parameters:
  $\phi \leftarrow \phi + \nabla[L_{\text{Scrub}} + L_{\text{ELBO}} + L_{\text{Recon}}]$
  $\theta \leftarrow \theta + \nabla[L_{\text{Recon}}]$
**until** convergence

---

**Scrubbing by adaptive quadratic discriminators (SC-VAE-QD)**

Consider the class-conditional Bayes classifier:

$$f_\psi(\mathbf{z}) = p(v = c|\mathbf{z}) \tag{9}$$

with likelihood:

$$p(z|v = c) = N(\mathbf{z}|\mu^c, \Sigma^c) \tag{10}$$

For multi-class problems, we maintain a *one vs. rest* estimator per class, $\psi = \{\mu^c, \Sigma^c, \mu^{c'}, \Sigma^{c'} \ \forall \ c \in C\}$. This estimator can be evaluated based on the Gaussian log-likelihood:

$$L(\mathbf{z}, v; \psi^c) = \ell(\mu^c, \Sigma^c|\mathbf{z}, v = c) + \ell(\mu^{c'}, \Sigma^{c'}|\mathbf{z}, v \neq c) \tag{11}$$

---

**Algorithm 2** SC-VAE-QD

---

Initialize network parameters
$\phi, \theta$
Initialize parameters of two classifers
$\psi_a, \psi_b$
$\lambda \leftarrow \alpha \overrightarrow{\mathbf{1}}_C, \ \alpha \in (0, 1)$
**repeat**
    Draw minibatch of $\{\mathbf{x_k}, \mathbf{v_k}, \mathbf{z_k}\}$ as in Alg 1
    $L_{\text{Scrub}} = 0$
    **for** $c \in C$ **do**
        Evaluate the Gaussian log-likelihood ratio for each classifier and average scrubbing loss:
        $L_{\text{Scrub}} = L_{\text{Scrub}} + \frac{1}{K}\sum_K L(\mathbf{z}_k, v_k; \psi^c)$
        Update class means and covariances for both estimators:
        $\mu^c = (1 - \alpha^c)\mathbb{E}_{\mathbf{v_k}=c}[\mathbf{z_k}] + \alpha^c \mu^c$
        $\Sigma^c = \text{Cov}_{\mathbf{v_k}=c}[\mathbf{z_k}, \mathbf{z_k}]$
        $\mu^{c'} = (1 - \alpha^{c'})\mathbb{E}_{\mathbf{v_k}\neq c}[\mathbf{z_k}] + \alpha^{c'} \mu^{c'}$
        $\Sigma^{c'} = \text{Cov}_{\mathbf{v_k}\neq c}[\mathbf{z_k}, \mathbf{z_k}]$
    **end for**
    Update network parameters as in Alg 1
**until** convergence

---

**Automatically tuning forgetting factor**

One of the core features of the moving average scrubbing methods (SC-VAE-MALS and -QD) is the automatic tuning of the forgetting factor, $\lambda$. Given a discriminator function, $f_\psi(\mathbf{z})$, and a minimization objective, $L(z, v; \psi)$, we simultaneously estimate two discriminators from the same function family, $f_{\psi_a}(\mathbf{z})$ and $f_{\psi_b}(\mathbf{z})$. The estimators use distinct forgetting factors, $\alpha_a$ and $\alpha_b$, with a fixed offset $\epsilon$. We tune both $\alpha_a$ and $\alpha_b$ in the direction of the better-performing discriminator.

---

**Algorithm 3** Automatically tuning forgetting factor ($\lambda$)

---

$\psi_a, \psi_b \leftarrow$ Initialize parameters of two discriminators
Initialize forgetting factors with fixed offset: $\epsilon$
$\alpha_a \in (0, 1 - \epsilon)$
$\alpha_b \leftarrow \alpha_a + \epsilon$
**repeat**
    Draw batch with $K$ samples:
    $(\mathbf{x_k}, \mathbf{v_k}), \mathbf{z_k} \sim q_\phi(\cdot \mid \mathbf{x_k}) \in \mathbb{R}^{D \times K}$
    Average the losses of the two discriminators to obtain the scrubbing loss
    $L_{\text{Scrub}} = \frac{1}{2}[L(\mathbf{z_k}, \mathbf{v_k}; \psi_a) + L(\mathbf{z_k}, \mathbf{v_k}; \psi_b)]$
    Forgetting factors step by $\Delta$ in the direction of the better decoder between $f_{\psi_a}$ and $f_{\psi_b}$
    **if** $L(\mathbf{z_k}, \mathbf{v_k}; \psi_a) > L(\mathbf{z_k}, \mathbf{v_k}; \psi_b)$ **then**
        $\alpha_a = max(\alpha_a - \Delta, 0)$
        $\alpha_b = \alpha_a + \epsilon$
    **else**
        $\alpha_b = min(\alpha_b + \Delta, 1)$
        $\alpha_a = \alpha_b - \epsilon$
    **end if**
    Continue with updating $\psi_a$, $\psi_b$, $\phi$, and $\theta$ as described in Algorithms 1 and 2.
**until** convergence

---

**Scrubbing by kernel density mutual information estimation (SC-VAE-MI)**

In the previous objectives, we used adversarial parametric functions $f_\psi(\cdot)$ as proxies to estimate and minimize the maximum predictive information in eq. (2). Here we consider a nonparametric estimator of the mutual information between $\mathbf{z}$ and $\mathbf{v}$ as an alternative scrubbing objective. The mutual information between these random variables is defined as:

$$I(\mathbf{z}, \mathbf{v}) = \mathbb{E}_{\mathbf{z}, \mathbf{v}} \left[ \log \frac{p(\mathbf{z}, \mathbf{v})}{p(\mathbf{v})p(\mathbf{z})} \right] = \mathbb{E}_{\mathbf{v}, \mathbf{z}} \left[ \log p(\mathbf{v}, \mathbf{z}) - \log p(\mathbf{v}) - \log p(\mathbf{z}) \right] \quad (12)$$

We approximate the densities of the joint and marginal distributions using a kernel density estimator, as done in prior work (e.g., Moon et al., 1995). Given a set of samples from the joint distribution, $\{\mathbf{z_j}, \mathbf{v_j}\}_{j=1}^{J}$, we approximate the joint density function as being proportional to:

$$p(\mathbf{v}, \mathbf{z}) \approx \hat{p}(\mathbf{v}, \mathbf{z}) \propto \sum_{j=1}^{J} k([\mathbf{z}, \mathbf{v}], [\mathbf{z}_j, \mathbf{v}_j]) \quad (13)$$

where $[\mathbf{z}, \mathbf{v}]$ denotes vector concatenation, and $k(\cdot, \cdot)$ denotes Gaussian kernel with a user-defined bandwidth parameter $h$:

$$k(\mathbf{x}, \mathbf{x}') = \exp\left( -\frac{||\mathbf{x} - \mathbf{x}'||^2}{2h} \right) \quad (14)$$

We approximate the marginal distributions over $\mathbf{z}$ and $\mathbf{v}$ in a similar fashion:

$$p(\mathbf{v}) \approx \hat{p}(\mathbf{v}) \propto \sum_{j=1}^{J} k(\mathbf{v}, \mathbf{v}_j) \quad (15)$$

$$p(\mathbf{z}) \approx \hat{p}(\mathbf{z}) \propto \sum_{j=1}^{J} k(\mathbf{z}, \mathbf{z}_j) \quad (16)$$

Therefore, we can approximate the term appearing inside the expectation of eq. (12) as:

$$\log \frac{p(\mathbf{z}, \mathbf{v})}{p(\mathbf{v})p(\mathbf{z})} \approx \log \sum_{j=1}^{J} k([\mathbf{z}, \mathbf{v}], [\mathbf{z}_j, \mathbf{v}_j]) - \log \sum_{j=1}^{J} k(\mathbf{v}, \mathbf{v}_j) - \log \sum_{j=1}^{J} k(\mathbf{z}, \mathbf{z}_j) + C \quad (17)$$

where $C$ is a constant that does not depend on the distribution of $\mathbf{z}$ and $\mathbf{v}$. Now, given a second batch of samples from the joint distribution $\{\tilde{\mathbf{z}}_k, \tilde{\mathbf{v}}_k\}_{k=1}^{K}$, we can approximate the expectation appearing in eq. (12) with the empirical mean. Thus, we obtain:

$$I(\mathbf{z}, \mathbf{v}) \approx \hat{I}(\mathbf{z}, \mathbf{v}) = \frac{1}{K} \sum_{k=1}^{K} \log \frac{\hat{p}(\tilde{\mathbf{z}}_k, \tilde{\mathbf{v}}_k)}{\hat{p}(\mathbf{v})\hat{p}(\mathbf{z})} \tag{18}$$

$$= \frac{1}{K} \sum_{k=1}^{K} \log \sum_{j=1}^{J} k([\tilde{\mathbf{z}}_\mathbf{k}, \tilde{\mathbf{v}}_\mathbf{k}], [\mathbf{z}_j, \mathbf{v}_j]) \tag{19}$$

$$- \frac{1}{K} \sum_{k=1}^{K} \log \sum_{j=1}^{J} k(\tilde{\mathbf{v}}_k, \mathbf{v}_j) - \frac{1}{K} \sum_{k=1}^{K} \log \sum_{j=1}^{J} k(\tilde{\mathbf{z}}_k, \mathbf{z}_j) + C \tag{20}$$

as our estimate of the mutual information (up to an additive constant $C$). Importantly, this estimator is differentiable with respect to $\tilde{\mathbf{z}}_1, \ldots, \tilde{\mathbf{z}}_K$, allowing us to compute gradients to update the network's learned representations. We caution that this estimator suffers a well-known curse of dimensionality and must be used with care. We expect it to work best when the VAE representation is low-dimensional and large batches of data (i.e., large $K$ and $J$) are used to form the estimate. The choice of bandwidth parameter in the Gaussian kernel function is also critical; smaller choices of $h$ will require larger batches of data to form a stable estimate.

---

**Algorithm 4** SC-VAE-MI

---

Initialize network parameters
$\phi, \theta$
Initialize bandwidth parameter
$h \in \mathbb{R}$
**repeat**
    Draw batch with $K$ samples
    $\{\mathbf{x_k}, \ \mathbf{v_k} \in \mathbb{R}^N, \ \mathbf{z_k} \sim q_\phi(\cdot \mid \mathbf{x_k}) \in \mathbb{R}^D\}_{\mathbf{k}=1}^K$
    Draw batch with $J$ samples
    $\{\mathbf{x_j}, \ \mathbf{v_j} \in \mathbb{R}^N, \ \mathbf{z_j} \sim q_\phi(\cdot \mid \mathbf{x_j}) \in \mathbb{R}^D\}_{\mathbf{j}=1}^J$
    Using eq. (18)
    $L_{\text{Scrub}} = \hat{I}(\mathbf{z}, \mathbf{v})$
    Update network parameters as in Alg 1
**until** convergence

---

# B EXPERIMENT DETAILS

## B.1 CONDITIONAL MOTION GENERATION

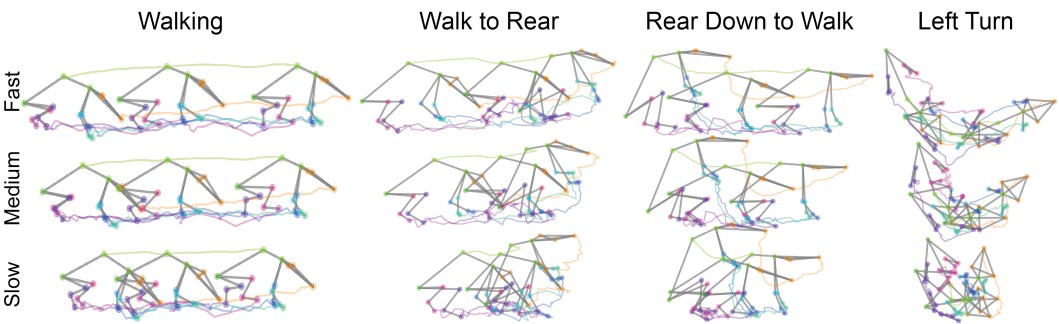

Figure 5: **Enabling conditional motion generation.** By manually adjusting the value of the conditioned variable, $\mathbf{v}_t$, we enable granular control over this variable in the generated output without changing the gross behavioral category. Here we change the conditioned speed and are able to adjust the speed of a behavior.

## B.2 LATENT VISUALIZATIONS

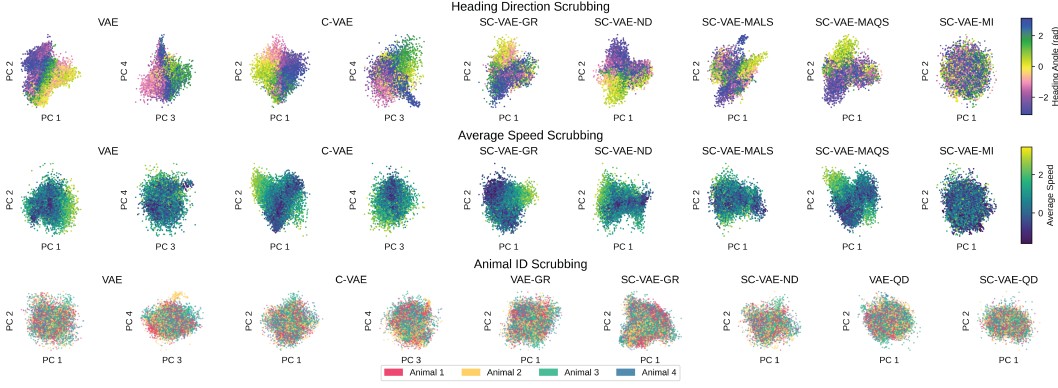

Figure 6: **PCA visualizations.** Projections of vanilla and scrubbed model representations to the top principal component dimensions, colored by heading *(top)*, speed *(mid)*, and identity *(bottom)*.

## B.3 OBTAINING BEHAVIORAL MOTIFS

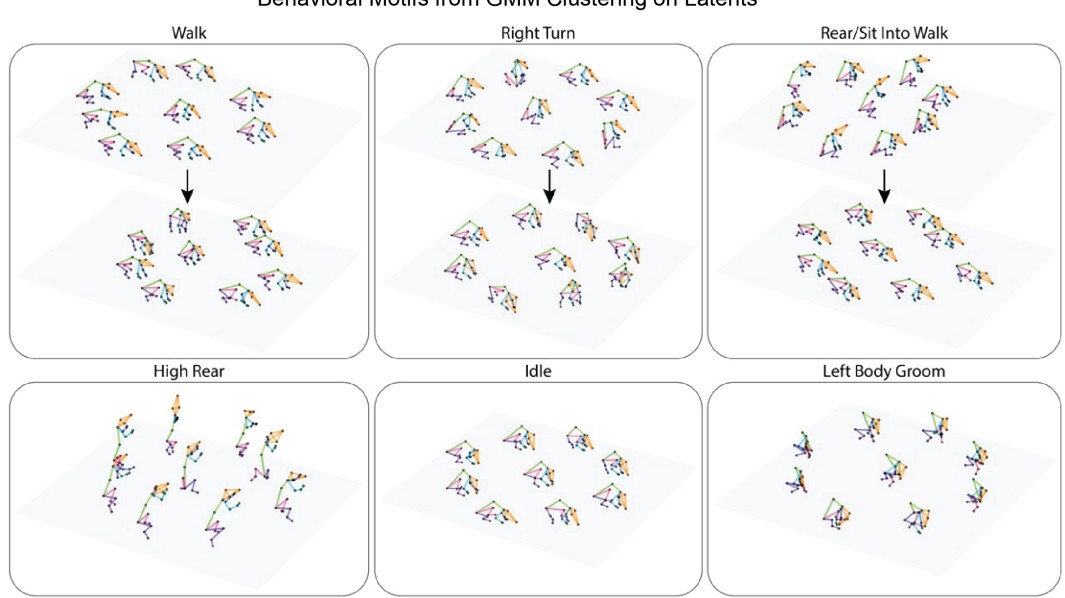

Figure 7: **Example motifs from GMM clustering.** Motifs extracted from our models' deep representations qualitatively reveal interpretable behavioral motifs.

## B.4 NUISANCE VARIABLE DETAILS

**Average speed** Average speed is a 3-dimensional vector corresponding to: (1) the average displacement of the root keypoint during a window (51 frames here), (2) the average speed of the spine and head relative to the root keypoint, and (3) the average speed of the limbs relative to the attached spine keypoint.

**Heading direction** Given a pose sequence, we calculate the yaw, $\alpha$, of the *mid-spine → front-spine* segment for the central frame of a pose sequence (i.e., the 26th frame of a 51 frame sequence in this paper). The final heading direction used by the scrubber is $[\sin \alpha \ \cos \alpha] \top$.

**Animal identity** We represent all categorical nuisance variables with one-hot encoding vectors.

## B.5 SCRUBBING IDENTITY IN VAE WITHOUT CONDITIONING

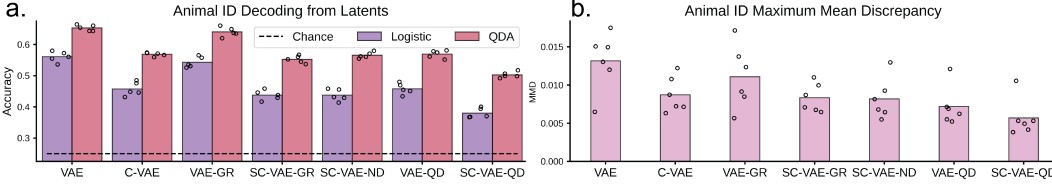

Figure 8: **Decoding identity from scrubbed latents.** (a) Classification accuracy for animal identity using logistic regression and quadratic discriminates. (b) Paired inter-individual MMD.

Given the need to maintain a reasonable reconstruction error when learning a scrubbed variational distribution, it follows that conditioning would be required when disentangling variables such as speed or heading. However, conditioning does not seem to be as intuitively necessary when disentangling variables, such as animal identity, which do not directly contribute to the reconstruction error. Despite

this, we find that conditioning with one-hot encodings on animal identity improves the scrubbing performance, as the linear and quadratic decodability of identity is reduced compared to scrubbing identity in a standard VAE (VAE-GR and VAE-QD in Fig. 8a). SC-VAE representations also reduced the paired maximum mean discrepancy (MMD) (Gretton et al., 2012) between individuals (Fig. 8b).

## B.6 METRIC DEFINITIONS

**One-component Gaussian log-likelihood** Given a mean-centered and standardized latent representation over a dataset, we split the walking behaviors $\mathbf{Z}^{walk}$ into $\mathbf{Z}^{walk}_{train}$ and $\mathbf{Z}^{walk}_{test}$. We use the mean $\boldsymbol{\mu}$ and diagonal covariance $\boldsymbol{\sigma}^2\mathbf{I}$ of $\mathbf{Z}^{walk}_{train}$ to calculate the Gaussian log-likelihood of $\mathbf{Z}^{walk}_{test}$:

$$\ln p(\mathbf{z} \mid \boldsymbol{\mu}, \boldsymbol{\sigma}^2\mathbf{I}) = -\frac{1}{2}\left[d\ln 2\pi + \ln|\boldsymbol{\sigma}^2\mathbf{I}| + (\mathbf{z} - \boldsymbol{\mu})\top(\boldsymbol{\sigma}^2\mathbf{I})^{-1}(\mathbf{z} - \boldsymbol{\mu})\right] \quad (21)$$

Thus, we quantify the level at which walking clusters of different speeds or heading directions can be represented by a single, merged Gaussian distribution. We cross-validated this over five folds.

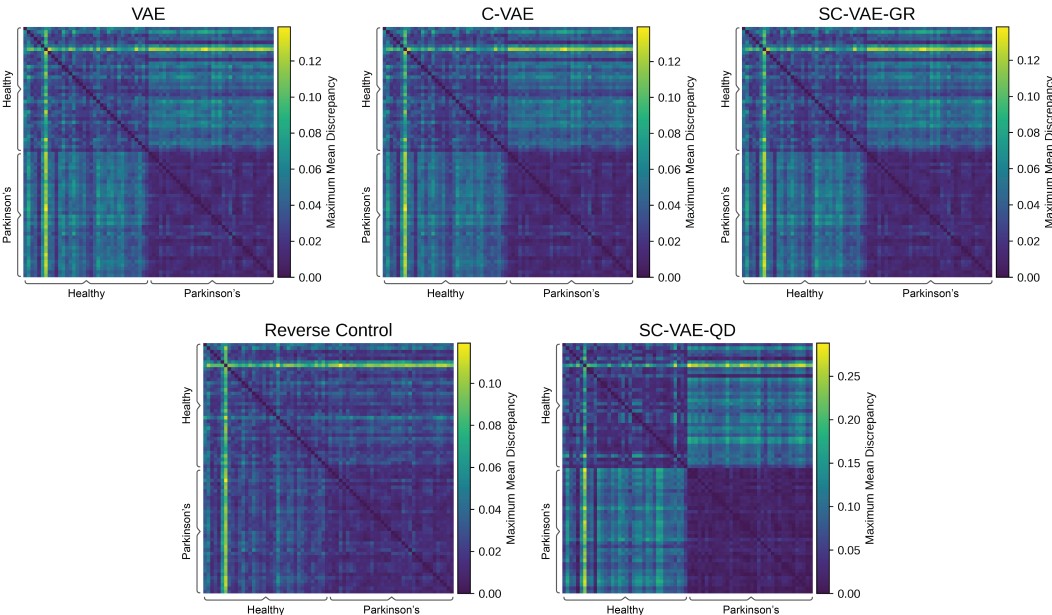

Figure 9: **Pairwise maximum mean discrepancy.** For each session (healthy or diseased) of each animal, we calculated the maximum mean discrepancy between their behavioral distribution. Qualitatively, the difference in the cross-condition MMD (*top-right* and *bottom-left* quadrants) vs. the in-condition MMD (*top-left* and *bottom-right* quadrants) is more pronounced in SC-VAE-QD compared to other models. This demonstrates the enhanced disease discriminability in our scrubbed models.

**Maximum mean discrepancy** We use maximum mean discrepancy (MMD) (Gretton et al., 2012) to define a distance between two latent behavioral distributions with $M$ and $N$ samples, $\mathbf{X} \in \mathbb{R}^{M \times D}$ and $\mathbf{Y} \in \mathbb{R}^{N \times D}$. Given a radial basis kernel function with bandwidth $h$:

$$k(\mathbf{z}, \mathbf{z}') = \exp\left(-\frac{||\mathbf{z} - \mathbf{z}'||^2}{h}\right) \quad (22)$$

MMD is defined as:

$$D_{MMD}(\mathbf{X}, \mathbf{Y}) = \frac{1}{M(M-1)}\sum_{i=1}^{M}\sum_{j\neq i}^{M}k(\mathbf{x}_i, \mathbf{x}_j) + \frac{1}{N(N-1)}\sum_{i=1}^{N}\sum_{j\neq i}^{N}k(\mathbf{y}_i, \mathbf{y}_j)$$

$$+ \frac{2}{MN}\sum_{i=1}^{M}\sum_{j=1}^{N}k(\mathbf{x}_i, \mathbf{y}_j) \quad (23)$$

Following the heuristic in Gretton et al. (2012), we define $h$ to be the median Euclidean distance in the aggregate sample $[\mathbf{X}, \mathbf{Y}]$. When visualizing the paired MMD between all available sessions (72 total) in Fig. 9, we qualitatively find increased differences across conditions compared to within conditions. We describe our quantification of this effect below.

In Sec. 4.5, we compare the healthy-disease discrepancy to the inter-individual discrepancy. Given a dataset of $I$ individuals where $\mathbf{H}_i$ is the healthy behavioral distribution and $\mathbf{D}_i$ is the diseased behavioral distribution of animal $i$, we calculate two metrics akin to statistical effect size:

$$[d_H]_i = \frac{(I-1) * D_{MMD}(\mathbf{H}_i, \mathbf{D}_i)}{\sum_{j \neq i} D_{MMD}(\mathbf{H}_i, \mathbf{H}_j)} \tag{24}$$

$$[d_D]_i = \frac{(I-1) * D_{MMD}(\mathbf{H}_i, \mathbf{D}_i)}{\sum_{j \neq i} D_{MMD}(\mathbf{D}_i, \mathbf{D}_j)} \tag{25}$$

Cohen's effect size is more similar to the cross-condition distance normalized to the variance within the control condition (healthy). Thus, we report $d_H$ in Table 2. Both metrics are reported in Table 3. Finally, we construct a k-nearest neighbor classifier ($k = 35$) to classify behavioral distributions using our MMD distance to quantify the changes in decoding accuracy of the disease label due to scrubbing.

Table 3: **Maximum mean discrepancy metrics.** We quantify the difference in an animal's cross-condition representations compared to the average in-condition difference.

| Model | Effect Size | |
|---|---|---|
| | $d_H \uparrow$ | $d_D \uparrow$ |
| VAE | $1.07 \pm 0.05$ | $2.69 \pm 0.20$ |
| C-VAE | $1.11 \pm 0.05$ | $2.82 \pm 0.22$ |
| SC-VAE-GR | $1.17 \pm 0.06$ | $3.07 \pm 0.24$ |
| SC-VAE-QD | $1.71 \pm 0.11$ | $5.55 \pm 0.44$ |
| Reverse Control | $0.68 \pm 0.03$ | $1.51 \pm 0.14$ |

## C  MODEL TRAINING DETAILS

### C.1  MODEL ARCHITECTURES

Table 4: **Residual block architecture.**

| Layer | Kernel | Stride | Padding | Activation | Normalization | Channels out |
|---|---|---|---|---|---|---|
| Input | - | - | - | None | None | in_channel |
| Conv1D 1 | 3 | 2 | 1 | PReLU | Batch | out_channel//2 |
| Conv1D 2 | 3 | 1 | 1 | None | None | out_channel |
| Conv1D skip | 3 | 2 | 1 | None | None | out_channel |
| Add | - | - | - | PReLU | Batch | out_channel |

Table 5: **Residual block transpose architecture.**

| Layer | Kernel | Stride | Padding | Activation | Normalization | Channels out |
|---|---|---|---|---|---|---|
| Input | - | - | - | None | None | in_channel |
| Conv1D$^\top$ 1 | 3 | 1 | 1 | PReLU | Batch | in_channel//2 |
| Conv1D$^\top$ 2 | 3 | 2 | 1 | None | None | out_channel |
| Upsample by 2 | - | - | - | None | None | out_channel |
| Conv1D skip | 4 | 1 | 1 | None | None | out_channel |
| Add | - | - | - | PReLU | Batch | out_channel |

Table 6: **Representation encoder architecture.**

| Layer | Kernel | Stride | Padding | Activation | Normalization | Output |
|---|---|---|---|---|---|---|
| Input | - | - | - | None | None | $51\times111$ |
| Conv1D | 7 | 1 | 3 | PReLU | None | $51\times128$ |
| Residual 1 | 3 | 2 | 1 | PReLU | Batch | $26\times256$ |
| Residual 2 | 3 | 2 | 1 | PReLU | Batch | $13\times512$ |
| Flatten | - | - | - | None | None | 6656 |
| Dense ($\mu$) | - | - | - | PReLU | None | 64 |
| Dense ($\sigma$) | - | - | - | PReLU | None | 64 |

Table 7: **Representation decoder architecture.** $c$ is the dimension of the disentanglement variable

| Layer | Kernel | Stride | Padding | Activation | Normalization | Output |
|---|---|---|---|---|---|---|
| Encoded input | - | - | - | None | None | $64 + c$ |
| Dense | - | - | - | PReLU | None | 6656 |
| Unflatten | - | - | - | None | None | $13{\times}512$ |
| Residual$^\top$ 1 | 3 | 2 | 1 | PReLU | Batch | $25{\times}256$ |
| Residual$^\top$ 2 | 3 | 2 | 1 | PReLU | Batch | $49{\times}128$ |
| Conv1D$^\top$ | 9 | 1 | 3 | Tanh | None | $51{\times}111$ |

Table 8: **Gradient reversal scrubber.** Uses ReLU activation for all dense layers. $c$ is the dimension of the disentanglement variable

| Layer | Input | Output |
|---|---|---|
| Gradient reversal (before MLP Ensemble) | - | - |
| MLP 1 | | |
| Dense 1 | 64 | 64 |
| Dense 2 | 64 | 64 |
| Dense 3 | 64 | $c$ |
| MLP 2 | | |
| Dense 1 | 64 | 64 |
| Dense 2 | 64 | $c$ |
| MLP 3 | | |
| Dense 1 | 64 | 64 |
| Dense 2 | 64 | 32 |
| Dense 3 | 32 | $c$ |
| MLP 4 | | |
| Dense 1 | 64 | 128 |
| Dense 2 | 128 | 128 |
| Dense 3 | 128 | $c$ |

### C.2 TRAINING DETAILS AND HYPERPARAMETERS

We standardized training such that the following are true across all models.

- Models were trained on NVIDIA A100 and V100 GPUs.
- Models were trained using AdamW optimizer (Loshchilov & Hutter, 2017) with a learning rate of 0.0001 and gradient clipping at 10,000.
- Losses for models had the same weights for joint position error and the VAE ELBO loss.
- Models were trained for 400 iterations over the training dataset with batch sizes of 2048.
- The model architecture described in Section C.1 was not changed across tasks.
- We used a pose sequence window size of 51 frames.
- The size of the latent dimension $z_t$ was 64 for the results in Sections 4.4-4.2 and 32 for the results in Section 4.5.

To ensure fair comparison across models, we limited hyperparameter-tuning to only method-relevant hyperparameters. General hyperparameters and optimizers were chosen to improve the joint position error in vanilla VAE reconstructions. These hyperparameters, as described above, were held constant across all VAE, C-VAE, and SC-VAE models.

Parameter tuning for SC-VAE scrubbers largely revolved around tuning the $\lambda$ of the $L_{\text{Scrub}}(\phi)$ loss. In most cases, increasing lambda increased the strength of the scrubber up to its linear or nonlinear assumptions. However, when $\lambda$ was too large, we found some cases of posterior collapse or mode collapse, primarily in the neural network scrubbing models (SC-VAE-GR and -ND). For SC-VAE-GR, we were additionally required to tune the scaling factor of the gradient reversal layer $\alpha$. For SC-VAE-ND, we found performance changes due to the number of discriminator training iterations in between

C-VAE iterations. If too few iterations were taken, the discriminator would underfit and vice versa. Due to our increment strategy for forgetting factors (Algorithm A.2), in training SC-VAE-MALS and SC-VAE-QD, $\lambda$ was the only required parameter to tune. In the polynomial extensions to the moving average algorithm (SC-VAE-MAQS), we found better performance with non-zero L2 regularization, $\beta$. Finally, for SC-VAE-MI, we additionally tuned the $h$ parameter determining the spherical width of the kernel.

Models were evaluated on a held-out validation set. Hyperparameters were selected based on the level of nonlinear invariance in the model latent representations to the scrubbed nuisance variable (i.e., the "MLP and "QDA" metrics in Fig. 2), filtering out models with posterior collapse. All reported metrics are calculated on a test set separate from the training and validation dataset. The results in Sections 4.4-4.2 were all calculated on the same set of models. Consequently, there were models not selected with better performance in the conditional motion synthesis task in Table 4.2. While increased invariance usually led to better performance in motion synthesis, we observed an inverse correlation in SC-VAE-ND models trained to scrub heading direction where the least invariant models had an $R^2$ of up to 0.859. Models were re-trained from scratch on the Parkinsonian dataset to obtain the results in Section 4.5.

Table 9: **Hyperparameter search.** We list the hyperparameters and values over which we trained models for our experiments. We applied grid search for methods with more than one hyperparameter.

| Model | Hyperparameters | Values |
|---|---|---|
| SC-VAE-GR | $\lambda$ | 10, 20, 50, 100 |
| | $\alpha$ | 0.8, 1.0, 1.2 |
| SC-VAE-ND | $\lambda$ | 10, 50, 100 |
| | $n\_iter$ | 1, 10, 50, 100 |
| SC-VAE-MALS | $\lambda$ | 10, 20, 50 |
| SC-VAE-MAQS | $\lambda$ | 5, 10, 20 |
| | $\beta$ | 0.25, 0.5, 0.75 |
| SC-VAE-MI | $\lambda$ | 250, 500, 750 |
| | $h$ | 0.25, 0.5, 0.75 |
| SC-VAE-QD | $\lambda$ | 5, 10, 20, 50 |

## D  PARKINSON'S DATASET

### D.1  IMPACT AND ANIMAL CARE STATEMENT

Phenotyping animal behavior is an early and essential step in drug discovery, and analyses of high-resolution 3D pose measurements provide a sensitive, reproducible, and scalable avenue for such preclinical assays. By releasing datasets and benchmarks for 3D pose analyses, we are supporting the continued development and acceleration of new phenotyping methods. We thus are facilitating the development of new therapeutics for neuropsychiatric disorders. All animal data were collected at AAALAC-accredited animal facilities. The care and experimental manipulation of all animals were reviewed and approved by Duke University Institutional Animal Care and Use Committee.

### D.2  LICENSES

Our dataset is released under a CC BY 4.0 Attribution International license, which permits sharing and adapting the datasets without restrictions except for providing attribution to the dataset when used or adapted in a new medium.

### D.3  DATASET SPLITS AND ACCESS

We used a 50-25-25 train-validation-test split for our datasets. All hour-long videos were split into 1-minute long sections and randomized across each dataset while ensuring equal distribution of animal identity. Additionally, as animal behavior can change across the length of a recording session, we ensured that each dataset received an even distribution for each third of the recording session. Our datasets, train-validation-test splits, Python code, and documentation for running all benchmarks can be found on our Github repository. See https://github.com/tdunnlab/scrubvae.

### D.4  ETHICS

Please see D.1 for a discussion of animal ethics. We acknowledge that by aiding the development of behavioral activity analysis approaches, bad actors could potentially use developed approaches for unethical purposes. However, we believe that the benefits behavioral analysis approaches offer for society, in terms of the potential to alleviate human disease burden, outweigh the (minimal) risks of misuse.

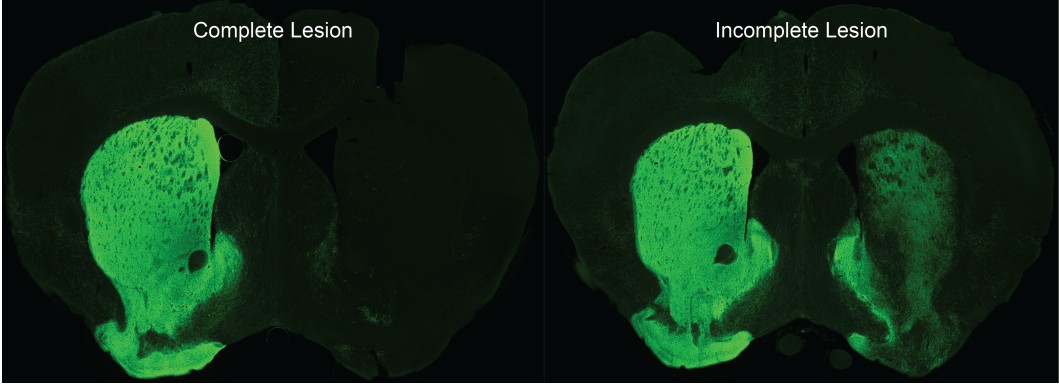

Figure 10: **Dopamine denervation in PD mouse brain.** Example fluorescent immunostaining for tyrosine hydroxylase (TH) in two mice after inducing Parkinson's disease via 6-hydroxy dopamine (6-OHDA) lesioning. TH (bright green) indicates the presence of dopamine neurons. *(left)* Example complete lesion. *(right)* Example incomplete lesion.

## D.5 DATASET COLLECTION

Adult Drd1a-Cre (8 females, 8 males, 18-20 weeks old) and Adora2a-Cre (10 females, 10 males, 18-20 weeks old) mice were anesthetized and implanted with a metal cannula (P1 tech) targeting the left medial forebrain bundle (from bregma: -0.7mm AP, -1.2mm ML, -4.75mm DV). Custom-built headbars were then affixed to the skull using UV glue and dental cement.

After allowing 1 week for the mice to recover, mice were habituated to head fixation and to the open field (30 cm x 30 cm) arena, then a 1-hour baseline recording was performed in the arena another 1 week later. Immediately following the 'baseline' session, mice were head-fixed, and 0.9 $\mu$L of a freshly made solution of 3.6mg/ml 6-hydroxy dopamine (6-OHDA) and 1mg/ml ascorbic acid was infused through the cranial cannula at a rate of 0.1 $\mu$L/min, allowing an additional 6 minutes for diffusion before removing the infusion cannula. 6-OHDA lesioned mice were provided with supportive care including daily subcutaneous saline injections and hand feeding with a mash made from a high-calorie diet supplement (Stat, ground-up macal) and ground up rodent feed (LabDiet 5001) until their weight stabilized. Two weeks after the lesion, mice were again placed in the open-field arena for 1 hour to collect their 'lesion' behavioral data.

At the conclusion of the experiment, mice were deeply anesthetized with isoflurane and underwent a transcardial perfusion with 15 mL of 0.1M phosphate buffered saline (PBS) followed by 50 mL of cold 4% paraformaldehyde (PFA) in 0.1M phosphate buffer (PB), pH 7.4 for fixation. Brains were then dissected from the skull and soaked overnight at $4^oC$ in the PFA solution, followed by 3 washes with 0.1M PBS. Each brain was then embedded in a mold in 5% agarose in order to collect 50 $\mu$M coronal sections with a vibratome (Leica VT1200S) into wells containing 0.1M PBS.

The next day, brain sections were immunostained for tyrosine hydroxylase (TH), a marker for dopamine neurons. Briefly, sections were washed in PBS before a 2-hour incubation in a blocking solution consisting of 5% goat serum, 3% bovine serum albumin, and 0.3% trition-x. Sections were then transferred to a half-block solution containing 1:1000 Rabbit $\alpha$-TH (PelFreez, P40101) overnight at $4^oC$ with agitation. Sections were then washed in 0.1M PBS containing 0.1% tween before a 4-hour incubation in a half-block solution containing 1:1000 goat $\alpha$-rabbit 488 (Invitrogen, A11008). Finally, sections were washed in PBS containing tween, then PBS prior to mounting on glass slides (VWR) and coverslipping with Vectashield Vibrance with DAPI (Vector Labs).

Fluorescent images of each immunostained coronal section were then collected on a VS200 slide scanner. Analysis of pixel intensity of the fluorescent signal for TH was performed using a custom MATLAB script. For each brain, 4 sections were analyzed that evenly span the anterior to posterior axis of the striatum. On each section, the striatum was manually segmented in both hemispheres, and the fluorescent signal was summed across the sections for both the healthy and lesioned hemispheres. The value for the fluorescence in the lesioned hemisphere was then divided by the value for the fluorescence in the healthy hemisphere to yield the 'integrated fluorescence' for each animal.

