# OpenReview forum: "Disentangling 3D Animal Pose Dynamics with Scrubbed Conditional Latent Variables"
_ICLR.cc/2025/Conference — ICLR 2025 Poster_

### Official Review · Reviewer_ZpEQ · 2024-10-31

**Soundness:** 2
**Presentation:** 3
**Contribution:** 2
**Rating:** 6
**Confidence:** 3

**Summary:**

The paper, "Disentangling 3D Pose Dynamics with Scrubbed Conditional Latent Variables," introduces the Scrubbed Conditional Variational Autoencoder (SC-VAE) as a framework for analyzing 3D pose data to extract behavior-relevant signals from nuisance factors like speed, direction, and individual traits. While the approach could offer valuable tools for behavioral research, several fundamental issues weaken the rigor, interpretability, and reproducibility of the work, limiting its potential impact.

**Strengths:**

The SC-VAE framework could provide useful tools in fields like neuroscience, where disentangling complex behaviors is essential. By addressing nuisance factors, SC-VAE theoretically enhances clustering and behavioral analysis. However, without stronger empirical validation, its practical contributions remain speculative.

**Weaknesses:**

-Key claims lack adequate empirical or theoretical support. For example, Line 52-53 states that "the other latent variables are not necessarily, nor typically, invariant to the conditional factor" without any justification or citation. Similarly, Line 60 lacks evidence to support the assertion that nuisance factors, like speed, do not impact behavior representation. This weakens confidence in the model’s purported advantages.

-Several statements are overly vague, such as Line 222-223: "if the latent dimension D is small, one expects the C-VAE to learn a disentangled representation." Without specifying a relative scale for 𝐷, it is difficult to understand or reproduce the model setup. Clearer definitions and justifications for parameters are necessary to enhance reproducibility and interoperability.

-The model is evaluated against only a single baseline, despite references to multiple adversarial models. This limited comparison restricts the ability to gauge SC-VAE’s robustness and leaves the evaluation incomplete, particularly given the lack of consideration for other adversarial baselines aimed at disentangling nuisance variables.

-Line 65 refers to "weak supervision" while the model receives the full ground truth for certain variables. This contradicts conventional definitions of weak supervision, potentially misleading readers regarding the model’s true level of supervision.

-The comparison between SC-VAE and C-VAE is not entirely fair, as SC-VAE benefits from direct optimization of latent variables, giving it an advantage. It would be fairer to include other adversarial methods in the comparison to better contextualize SC-VAE’s performance.

-Key implementation details, such as model architecture, hyperparameters, and balancing of loss components (particularly λ for term weighting), are missing. Without these, it is challenging to reproduce the results or determine if they arise from design choices or hyperparameter tuning.

-Visualizations referenced in Line 397 are cluttered and lack clear captions, making them difficult to interpret and detracting from the study’s overall clarity. Simplifying these figures and providing self-explanatory captions would improve readability and support the model’s claims more effectively.

-The log-likelihood (LL) calculation presented in Figure 3.C is not well explained, making it hard to interpret the metric’s relevance to the conclusions. A more detailed description of the calculation process would improve transparency.

-The authors claim SC-VAE can extend to other species and behaviors, but no additional experiments support this assertion. Expanding validation to other datasets is crucial for establishing generalizability.



In summary, the paper presents a potentially valuable framework for disentangling behavior-relevant signals from nuisance factors in 3D pose data. However, critical gaps in methodology, inadequate baseline comparisons, and insufficient experimental details limit the work’s robustness and reliability. Addressing these issues—through clearer claims, more thorough benchmarks, and comprehensive experimental details—would be essential for making a strong contribution to the field. As it stands, the paper falls short of the standards necessary for acceptance in its current form.

**Questions:**

What evidence supports the claim that nuisance factors like speed do not affect behavior representation (Line 60)? Has this assumption been empirically tested?

Could you offer more insight into how parameter choices, especially the selection of λ for weighting loss terms, affect the model's performance?

Given that SC-VAE optimizes latent variables directly, would you consider including other adversarial methods for a more balanced comparison? How might this impact the conclusions drawn about SC-VAE’s performance?

Would you consider adding comparisons with other adversarial baselines designed to disentangle nuisance variables? This would provide a clearer assessment of SC-VAE's robustness in comparison to similar methods.

**Details Of Ethics Concerns:**

Data from experiments involving rats generally requires ethical review and approval.

---

> ### Author Response · Authors · 2024-11-19
> **Initial author rebuttal (1)**
>
> We thank the reviewer for their thoughtful insights. We are glad the reviewer recognizes the importance of our research directions in neuroscience. Some of the comments in this review will take some time to address. However, we would like to submit this initial rebuttal to initiate discussion. By the end of the author-reviewer discussion period, all comments will be fully addressed. We thank the reviewer for helping us improve our paper.
>
> > Key claims lack adequate empirical or theoretical support. For example, L52-53…
>
> We understand that the language was confusing here. The evidence of the claim in L52-53 is shown in the results in Figure 2A-D. In these panels, we show that C-VAE latent variables are not invariant to the conditioned factor and contribute significantly to the latent structure. We will clarify this language.
>
> > Similarly, Line 60 lacks evidence...
>
> To clarify, we assert the opposite: “certain nuisance variables, like speed, are **not** always completely independent of meaningful behavioral representations”. Our results in Fig 3F-I demonstrate this. When complete independence is assumed when not the case, as in SC-VAE-MI, performance decreases compared to models with more relaxed assumptions (SC-VAE-MALS). Furthermore, we did not intend to argue that speed is always a nuisance variable. However, in many cases, variables like speed can lead to over-segmentation in an action space and less interpretable representations (see [1]).
>
> > Several statements are overly vague …
>
> We thank the reviewer for this suggestion. We will expand our Appendix B to include more justifications behind our parameter choices.
>
> > The model is evaluated against only a single baseline, despite references to multiple adversarial models…
>
> In case of misunderstanding, we would like to highlight that the SC-VAE-GR model in our paper is a baseline using an existing adversarial method [2]. Furthermore, we intended one of our primary contributions to be defining explicit functional forms for the scrubbers. This bypasses the need to train and tune separate adversarial neural networks (ANN) as existing methods do. In Fig 3, we show the benefits of using explicit functions instead of nondescript ANNs for scrubbing.
>
> > Line 65 refers to "weak supervision"...
>
> We understand that the term weak supervision can be vague. We based our definitions on those in papers like [3] in which our approach falls into the category of “restricted labeling”. A fully supervised approach would use action class labels during training. However, we are open to changing our language to something like “partial supervision” to make it clear that we are supervising with a subset of features.
>
> > The comparison between SC-VAE and C-VAE is not entirely fair…
>
> Indeed, many current methods of disentanglement in behavioral neuroscience do not apply direct optimization similar to C-VAE [4-6]. We introduced this comparison exactly to illustrate the advantage of doing so. As explained above, we also compared to a different baseline where there is direct optimization of latent variables (SC-VAE-GR).
>
> [1] Costacurta, J. et al. Neurips (2022). [2] Ding, Z. et al. CVPR (2020). [3] Shu, R. et al. ICLR (2020) [4] Whiteway, M.R. et al. PLoS Comp Bio (2021). [5] Shi, C. et al. Neurips (2021) [6] Yi, D. et al. eLife (2023)

---

> > ### Author Response · Authors · 2024-11-19
> > **Initial author rebuttal (2)**
> >
> > > Key implementation details … are missing…
> >
> > Section B of the Appendix contains additional details. The only hyperparameter, $\lambda$, was tuned using cross-validation. We will expand this appendix section to be as exhaustive as possible, and we are working on releasing the code on GitHub as part of the review process. We will also include more details and references to Appendix B in the main text.
> >
> > > Visualizations referenced in Line 397 are cluttered and lack clear captions…
> >
> > We thank the reviewer for this suggestion. We will adjust the presentation of our figures in the final PDF.
> >
> > > The log-likelihood calculation presented in Figure 3C is not well explained…
> >
> > We agree and will add detailed explanations for all calculated metrics. To quantify the level at which walking behaviors can be merged into a single Gaussian cluster, we are using the $\mu$ and $\Sigma$ of $\mathbf{z}$ for a subset of walking behaviors to calculate the Gaussian log-likelihood of the rest of the walking behaviors. We applied this over five folds for cross-validation.
> >
> > > The authors claim SC-VAE can extend to other species and behaviors…
> >
> > We did not mean to imply any broad claims or specific predictions regarding how the model will perform on other species; although we expect it will perform reasonably since pose tracks will have similar structure and statistics. We will update our conclusion to emphasize future work in this direction. If there are specific lines in the text that the reviewer finds problematic, we would appreciate knowing the line numbers so that we can best address this concern.
> >
> > > ...ethical review and approval.
> >
> > We thank the reviewer for bringing up this concern. The studies were approved by the university IACUC. We will include an `Impact and Animal Care` and  `Ethics` statement in the Appendix.
> >
> > > How might [other adversarial baselines] impact the conclusions drawn about SC-VAE’s performance?
> >
> > This is a great question. We can include additional baselines. However, first, this paper’s contribution is to introduce ideas of adversarial disentanglement to behavioral neuroscience analysis. These methods are novel and essential to this application. Second, as you observed, nuisance variables may affect behavioral representations in distinct ways. Thus, one of our additional technical contributions is to introduce explicit functional definitions for the scrubber which enhances control over the level of disentanglement. In Fig 2A-C of our paper, we show that while the neural network scrubber (SC-VAE-GR) has nonlinear assumptions of disentanglement, it only achieves linear disentanglement. We believe these are novel and significant contributions which would not be lessened by additional baselines.

---

> > > ### Comment · Reviewer_ZpEQ · 2024-11-25
> > >
> > > I want to thank the authors for their efforts in addressing my comments. However, some of their explanations are not directly evident in the text and require further clarification to enhance readability. For instance, while the authors state, *"Furthermore, we intended one of our primary contributions to be defining explicit functional forms for the scrubbers,"* the motivation and reasoning behind the specific definition of these functional forms are not clearly articulated in the manuscript. At the very least, an ablation study would be necessary to demonstrate the importance of each component within the scrubbers and their contribution to the overall model. While the authors have addressed some of my concerns, I will revise my rating to 5.

---

> > > > ### Author Response · Authors · 2024-11-27
> > > > **Following up with a revised PDF**
> > > >
> > > > We thank the reviewer for their engagement and follow-up questions. We have uploaded a revised version of the paper and believe it is significantly improved thanks to the discussion here. Please see the global response to all reviewers for a full summary of changes. We note specific changes in response to the reviewer here.
> > > >
> > > > > Additional baseline
> > > >
> > > > In addition to our previous VAE, C-VAE, and SC-VAE-GR baselines, we have added a new baseline, SC-VAE-ND. We have also reformatted our former Sec 3.4 to better differentiate these baselines from our new approaches (SC-VAE-MALS, -MAQS, -QD, and -MI) and dedicate a section to the challenges associated with these baseline adversarial disentanglement methods. Please see the updated Sections 3.3-3.5.
> > > >
> > > > > Fixing clarity in response to comments by the reviewer
> > > > * We have updated L52-53 to say “We will show …”. We also re-state this in L345-346 closer to the relevant result.
> > > > * We have rephrased the former L60 to “nuisance variables, like speed, have **some** dependencies with action understanding” for better clarity.
> > > > * We have removed all references calling our method weak supervision.
> > > > * We have enlarged and moved the former Figure 2e to the Appendix and updated the caption.
> > > > * In Appendix B.6, we used equations to define the log-likelihood calculation in Fig 3c, f *bottom*.
> > > >
> > > > > Expanded Appendix
> > > >
> > > > We followed the reviewer’s suggestion to expand the appendix. We now have full details of model architectures, training, hyperparameter tuning, and our datasets (including ethics statements). We have also included clarifying information about our experiments and methods. If the reviewer feels anything is still missing, we will be happy to include it in the camera-ready version.
> > > >
> > > > > the motivation and reasoning behind the specific definition of these functional forms are not clearly articulated in the manuscript
> > > >
> > > > We thank the reviewer for this suggestion. To improve clarity on these motivations, we have made the following changes to the text of our revised PDF.
> > > > * L75-78 now better clarify the challenges we are seeking to overcome.
> > > > * We believe our reformatted Sec 3.4-3.5 will help with this motivation. Particularly, see the new “Challenges” section L270-280.
> > > > * Added titles “Linear”, “Categorical”, and “Nonlinear” to the subsection titles in Sec 3.6 to denote the purpose of each scrubber.
> > > > * We have added a footnote in L376-377 which discusses intuition for different types of scrubbing for different variables.
> > > > * Added phrasing "These results suggest the need for disentanglement methods with weaker (-MALS) and stronger (-MI) assumptions for ..." in L477-478.
> > > >
> > > > > At the very least, an ablation study would be necessary to demonstrate the importance of each component within the scrubbers and their contribution to the overall model.
> > > >
> > > > We agree that this is important. One thing we can add to the camera-ready version would be an exploration of how our automatic smoothing factor tuning algorithm affects results. Otherwise, we appreciate the opportunity to highlight that our scrubbers are simple functions (e.g., linear, quadratic, etc.) that do not contain multiple components to ablate.
> > > >
> > > > We would also like to highlight our existing ablation comparisons: 1) VAE vs C-VAE, 2) neural network scrubbing (SC-VAE-GR, -ND) vs parametric scrubbing (SC-VAE-MALS, -MAQS, -QD, -MI), 3) linear (-MALS) vs nonlinear scrubbing (-MI), and 3) scrubbing in C-VAE vs VAE.
> > > >
> > > > We thank the reviewer again for their involvement in this process and appreciate the opportunity to improve our paper. If the reviewer feels anything is insufficiently addressed, please let us know.

---

> > > > > ### Comment · Reviewer_ZpEQ · 2024-12-02
> > > > >
> > > > > Thank you to the authors for their detailed response. It addresses my major concerns, and I have adjusted my score to 6.

---

### Official Review · Reviewer_qJjc · 2024-11-03

**Soundness:** 3
**Presentation:** 3
**Contribution:** 3
**Rating:** 8
**Confidence:** 4

**Summary:**

The authors propose a framework for modeling 3D pose dynamics which is capable of disentangling both continuous and discrete nuisance factors (such as speed and animal identity). They propose to use a conditional variational autoencoder, where the conditioning variables are the nuisance factors to disentangle. In order to properly ensure disentanglement of the nuisance factors from the VAE latents, the authors then propose a series of strategies for "scrubbing" the nuisance factors from the latents. They then demonstrate the effectiveness of these various strategies using 3D poses from a mouse behavioral experiment in a series of well-designed control studies. Finally, they apply these strategies to a larger dataset of healthy and diseased mice and demonstrate their ability to accurately discriminate between these two groups.

**Strengths:**

the problem is well-motivated appropriate for the ICLR audience, and the paper is generally well-written

the solution proposed by the authors is a creative synthesis of the existing literature, as well as some extensions necessary to make their solution work in practice

the validation experiments shown in figs 2a-c are strong, with appropriate baselines (I like the use of VAE Processed as a baseline in the heading direction experiments)

consistent motion synthesis (sec 4.3) is also a neat way to test disentangling

**Weaknesses:**

the authors test a wide range of models, but there is a lack of comparison to other established disentangling methods. while this literature is large, it would, for example, be useful to compare speed disentangling against the model in costacurta et al 2022. this model builds invariance to a specific nuisance variable (speed) and was designed for this exact type of data; if the authors could show that one of their models performs on par with the costacurta model, and can disentangle other generic variables besides, it would strengthen the argument for the scrubbing approach with pose data.

some details of the experiments are not clear. in sec 4.2, is there a separate model fit for each of the nuisance variables? if so, what happens if a single model is trained to scrub both speed and heading direction? in general, it is not clear if this approach is robust to scrubbing multiple variables at once.

the MMD analysis in sec 4.5 should be expanded upon, at the moment it is hard to understand. A simple decoding analysis (healthy vs diseased) would be a nice complement, and could lead to a punchier conclusion ("we could decode disease state x% of the time in the VAE, and y% of the time in the SC-VAE-QD"). I am also somewhat surprised that scrubbing leads to _improved_ discriminability between healthy and diseased conditions; I suppose this is because there are systematic differences between the two conditions that are obscured by animal identity? There are many such healthy vs diseased experiments with transgenic lines where a single subject will fall into one or the other category, but never both. In this case _not_ scrubbing subject ID would likely lead to higher discriminability (though perhaps for uninteresting reasons, such as diseased subjects are slightly smaller on average, etc.)

unsupervised clustering of pose data is becoming more ubiquitous in large-scale drug screens and disease research. the differences between two disease models, or the effects of a drug, can manifest in subtle differences in animal behavior. one of the potential drawbacks of this approach is that, by scrubbing certain information from the latent representation, these subtle differences may also be lost. while I think this work is super interesting and useful, I think a fuller discussion of its potential drawbacks (and possibly how these can be controlled) should be included in the Conclusions section.

**Questions:**

the sentence starting on L76 is very dense and difficult to parse: "we show that simple linear estimators perform favorably to traditional neural adversarial paradigms while introducing strategies for nonlinear estimators which bypass the need for specifying adversarial neural networks and their hyperparameters." can this be clarified?

fig 1 contains a lot of information but is difficult to see at its current size (especially 1c), can this be enlarged?

L293: "we increment or decrement the forgetting factors based on which filter provides a better fit to the minibatch statistic" - this seems to imply either/both forgetting factors can change, which contradicts the previous statement that one factor is "fixed" - please clarify

sec 4.4: the authors find that different scrubbing strategies are appropriate for heading vs speed. This is an interesting finding, but what does it mean for the practical applicability of this method?

do the authors have any intuition for what features are being removed by scrubbing subject identity? is it just subject size, or something else? if the 3D poses were normalized within each subject by, say, distance from tailbase to neck (or some other such pair of points), does the subject id scrubbing still result in disentangling?

as a control experiment, if the authors scrubbed disease state instead of animal identity in the final analysis, does the MMD drop to a chance level?

---

> ### Author Response · Authors · 2024-11-18
> **Initial author rebuttal (1)**
>
> We thank the reviewer for their thorough and thoughtful comments. We are glad the reviewer found our work appropriate for the ICLR audience and recognized our efforts to increase the practicality of learning disentangled representations in neurobehavioral data. Some of the comments in this review will take some time to address. However, we would like to submit this initial rebuttal to initiate discussion. By the end of this discussion period, all comments will be fully addressed. We thank the reviewer for helping us improve our paper.
>
> > On Costacurta et al. 2022 …
>
> Despite similar motivations between our work and Costacurta et al. 2022, we found some challenges that made comparison nontrivial. First, their method was developed for depth camera recordings, whereas we used 3D keypoint trajectories. The adaptation to another modality presents a nontrivial challenge (indeed there was an entire Nature Methods paper published on this [1]). Second, their method is a time series clustering method, whereas our goal was to learn a low-dimensional feature space useful for many downstream tasks beyond clustering. Furthermore, our methods are complementary, not mutually exclusive. In principle, the methods developed in Costacurta et al. 2022 can be used to cluster our low-dimensional representation similar to [2]. Finally, as you have already stated, our goal was to allow flexibility for disentangling kinematics beyond speed.
>
> That being said, we agree that baselines are important. We would like to emphasize that we include three baselines that correspond to many methods in the literature: VAE, C-VAE, and SC-VAE-GR. In the context of behavioral disentanglement, VAE corresponds to [2]; C-VAE to [3-5]; SC-VAE-GR to [6].
>
> > Is there a separate model fit for each of the nuisance variables …
>
> For the consistency of comparison in this paper, we trained models from scratch and separate models were fit for each of the nuisance variable scrubbers. We can include some preliminary results in the Appendix from models trained on scrubbing both heading direction and average speed which show more pronounced benefits due to scrubbing. It will be interesting in future work to explore scrubbing with inter-dependencies between nuisance variables in more detail. For example, heading direction and average speed are not completely independent in our dataset. This can be seen in (Fig. 3D, right) where there are four heading direction peaks within walking behaviors. This is because mice tend to walk along the four walls of the arena in our dataset.
>
> > The MMD analysis…
>
> We will rewrite this section to make our motivation more clear. Previous works have used MMD to quantify distances in VAE latent spaces, a notable example in neuroscience is [7]. In our analyses, the MMD ratio is similar to a measured effect size between healthy and diseased conditions. It captures the size of the difference between behavioral distributions relative to the within-condition differences across animals. When the variability across animals decreases (when animal identity is scrubbed), we gain a more precise estimate of the disease effect. We chose to use MMD over alternative measures of distribution difference, e.g., Jensen-Shannon Divergence, because MMD captures differences without discretization.
>
> We think our current MMD analysis is a simple proof of principle that illustrates a potential benefit of our approach while being rooted in a metric used by prior work in the field. Rather than affecting binary healthy vs. diseased classification, increased separation between these two conditions (as quantified by MMD) is better considered as a way to reveal subtler phenotypic differences that would otherwise be obscured by animal-animal variability, and to potentially increase sensitivity for detecting the effects of additional experimental manipulations (e.g., optogenetics, ablations, pharmaceuticals, or therapies). Our partial lesion analyses (Fig 4B) are an example of this.
>
> > …transgenic lines where a single subject will fall into one or the other category…
>
> Here our identity scrubbing paradigm makes sense for our dataset in which we have recordings of every subject both before and after inducing PD symptoms. However, as you say in your described scenario with unpaired data, full identity scrubbing may be less appropriate. In that case, SC-VAE can be used in principle to scrub identity only within each condition to enhance the disease vs healthy effect size.
>
> [1] Weinreb, C. et al. Nature Methods (2024) [2] Luxem, K. et al. Nature Comm (2022) [3] Whiteway, M.R. et al. PLoS Comp Bio (2021) [4] Shi, C. et al. Neurips (2021) [5] Yi, D. et al. eLife (2023) [6] Ding, Z. et al. CVPR (2020) [7] Goffinet, J. et al. eLife (2021)

---

> > ### Author Response · Authors · 2024-11-18
> > **Initial author rebuttal (2)**
> >
> > > Unsupervised clustering of pose data is becoming more ubiquitous…
> >
> > Importantly, information about heading direction, speed, etc. is never lost from the latent representation in our approach, it is placed into orthogonal dimensions (i.e. “disentangled” from the other latent factors).
> >
> > > I think a fuller discussion of its potential drawbacks...
> >
> > We agree and will expand our discussion of limitations. For example, while we believe our method produces latent spaces that are more interpretable and produce higher quality clusters, it is difficult to quantitatively benchmark and verify this. Additionally, nuisance factors are not always known *a priori* and may be difficult to compute based on raw pose features. These limitations broadly apply to current methods, including our own. We believe our flexible approach allowing for weaker (linear) and stronger (nonlinear) scrubbing can help practitioners navigate some of these challenges and tradeoffs.
> >
> > > L76 is difficult to parse:
> >
> > One of our primary contributions is to use explicitly defined functions for the scrubbers (SC-VAE-MALS, -MI, etc.) instead of traditional adversarial neural network approaches (SC-VAE-GR). Our methods do not require the specification of a second neural network architecture or the tuning of associated hyperparameters. Our results show that even our simplest linear scrubbers (SC-VAE-MALS) perform favorably to neural network scrubbers (SC-VAE-GR).
> >
> > > Can Fig 1 be enlarged?
> >
> > Yes, we will incorporate this suggestion along with the figure changes suggested by the other reviewers in our final PDF.
> >
> > > L293 …
> >
> > The difference between the factors is fixed, not the factors themselves. In other words, you can imagine that the two factors represent a fixed-width interval that is automatically shifted until it is centered around the ideal factor. We see how this is unclear in the text and will clarify it with specific references to Sec A3 in the appendix.
> >
> > > Sec 4.4: the authors find that different scrubbing strategies are appropriate for heading vs speed…
> >
> > We believe that this enhances the practicality of adversarial disentanglement methods. As described above, adversarial disentanglement methods typically use nondescript neural networks for the “scrubber”. Here we have shown that the explicit functional form of the disentanglement matters.
> >
> > > is it just subject size…
> >
> > To clarify, our rotation representation does not provide any direct notion of the subject size or segment length. However, subject size cannot be trivially preprocessed. For example, normalizing for total size would not normalize relative limb proportions. Animals with different limb proportions would naturally have different postures and kinematics across behaviors. These differences would nonlinearly affect dynamics even in rotation representations without explicit segment length information.

---

> > ### Comment · Reviewer_qJjc · 2024-11-25
> >
> > I thank the authors for their response. With regards to fitting a separate model for each nuisance variable, it could be interesting to include some preliminary results in the Appendix, but the more important thing from my point of view is to make it clear that the models are in fact only scrubbing one nuisance variable at a time, and what the pros/cons of attempting to scrub multiple variables simultaneously at once would be.
> >
> > The MMD description given in this rebuttal is much clearer than in the submission, so I think just expanding on what the metric is and why it was chosen in the main text would be very helpful.

---

> > > ### Author Response · Authors · 2024-11-27
> > > **Following up with a revised PDF**
> > >
> > > We thank the reviewer for the discussion. We have uploaded a revised version of the manuscript. Thanks to the reviewer’s suggestions, we believe that the paper has improved significantly. Please see the global response for a full summary of changes. We list the changes specific to this review here.
> > >
> > > > Additional baselines
> > >
> > > Following the reviewer’s suggestions, we have included a new adversarial disentanglement baseline (SC-VAE-ND). Please see the response to all authors for more details. We thank the reviewer for their feedback.
> > >
> > > > Improved Parkinson’s analysis (Sec 4.5)
> > >
> > > We have also revamped the Parkinson’s analysis section to better communicate the motivation and intuition behind our metrics.
> > > * We have rephrased our language to clarify our intuition behind the MMD metrics in writing in the main text. Further details and equations have been added to Appendix B.6.
> > > * We have converted the results from a figure to a table to enhance the presentation.
> > > * For conciseness in the main text, we have moved the disease in-condition effect size to the appendix in Table 3. The healthy in-condition effect size remains and has been rephrased for clarity.
> > > * To further enhance the reader’s intuition, we have presented a heatmap of pairwise MMD between all behavioral sessions in Fig. 9. We believe the effect of scrubbing on disease discriminability is visually evident in this figure.
> > > * As suggested by the reviewer, we have added an accuracy metric using a k-nearest neighbor classifier to Table 2 **middle**.
> > > * The reviewer also had a great suggestion to add a scrubbing control where the disease label itself is scrubbed instead of the identity. We have added this new baseline (see Reverse Control).
> > >
> > > > Text and figure improvements
> > >
> > > * We have rephrased the former L76 of our introduction. Please see the updates in L75-78 of our introduction.
> > > * To give more intuition on animal identity and subject size effects, we have added some motivating sentences in L284-287 to help with this intuition.
> > > * We have expanded our limitations section to include discussions of 1) multi-variable scrubbing and 2) the drawbacks of scrubbing in unknown datasets.
> > > * In L253-254, we now explicitly state that we are scrubbing variables one at a time and give our rationale for doing this.
> > > * We have updated and enlarged all elements of Fig 1. Instead of the schematic formerly in Fig 1a, we show real low-dimensional projections originally from former Fig 2e. Former Fig 1c has been moved to Appendix Fig. 5.
> > > * We have rephrased the explanation of forgetting factors now in L288-289.
> > >
> > > We thank the reviewer again for their role in improving our paper. Let us know if there are any further questions.

---

> > > > ### Comment · Reviewer_qJjc · 2024-12-02
> > > >
> > > > I thank the authors for their extensive improvements. The Parkinson's analysis is indeed much clearer now, and the other changes have improved the manuscript. I have updated my score to an 8.

---

### Official Review · Reviewer_p9qK · 2024-11-03

**Soundness:** 3
**Presentation:** 2
**Contribution:** 3
**Rating:** 6
**Confidence:** 3

**Summary:**

The authors propose scrubbed conditional variational autoencoder (SC-VAE), a novel framework for disentangling nuisance factors by removing variable information from latent spaces by using an adversarial learning objective. They demonstrate the utility of SC-VAE using 2 mouse behavior datasets.

**Strengths:**

- This framework extends C-VAEs into a more interpretable realm by introducing scrubbing. The idea of scrubbing the nuisance variables out of data can allow researchers to focus on the variable of interest without the unnecessary/irrelevant information.
- The idea of isolating the semantic information from the character of the action has the potential to be applied to many fields other than neuroscience such as speech recognition or emotion classification.
- Clear flow throughout the paper.

**Weaknesses:**

- Generalizability: Since there is the need for specific assumptions and constraints to guarantee disentanglement, e.g.  picking the known factors, SC-VAE might be inflexible across datasets with unknown structures. I know that the unsupervised methods were briefly mentioned at the beginning and the most prominent weakness to them seemed to be the sensitivity to nuisance variability but this then brings the flexibility in mind once again in terms of more complex datasets with less prior knowledge on the structure. I would be interested to know how SC-VAE could be adapted to work in cases where nuisance factors are hard to determine.
- Figures: Figures can be more polished. Right now many of them look like default matplotlib figures (See minor points for improvement suggestions).

Minor points

- For each plot in Fig 2.a,b,c,d, the top and rightmost axis can be removed, and for Fig2.a.b the legend doesn't need to be repeated. Same for fig4.
- Fig3.f and fig3.i have overlapping histograms, and in general the axis labels are illegible in fig3.
- l398 there might be extra space in ‘Fig. 2e’.
- Fig4.a the second plot y axis label is too close to the title.

**Questions:**

- In terms of pose estimation, I often hear about DeepLabCut. I know that framework doesn’t use a VAE backbone but in general how does SC-VAE compare to DeepLabCut?

---

> ### Author Response · Authors · 2024-11-18
> **Initial author rebuttal**
>
> We thank the reviewer for their consideration and suggestions. We are glad that the reviewer recognized our motivations for more interpretable and researcher-centric methods of deep behavioral analysis, as well as the applicability of some of our ideas to other domains.
>
> > Generalizability: ….
>
> This is a very thoughtful point, and we agree that this is essential to consider. First, as the reviewer observed, works like Locatello et. al. 2019b have shown that unsupervised disentanglement is difficult, if not impossible, to guarantee. Much of the disentanglement literature has since moved towards introducing assumptions or weak supervision as we do (Shu et al. 2019; “Weakly supervised disentanglement with guarantees”). In our paper, our fully unsupervised baseline (VAE) consistently performs less favorably when compared to the SC-VAE models.
>
> Second, we would like to clarify that our intention was not to completely replace unsupervised behavioral representation learning. Our goal was to introduce approaches that complement an exploratory analysis of complex behavioral datasets. Here our nuisance factors (speed, heading, and identity) are easy to access in practice for most datasets. In Fig. 4, we show that by reducing the structure of nuisance identity effects, scrubbing enhances the subtle structure about PD itself (lesion severity).
>
> > Figures …
>
> We thank the reviewer for these suggestions for improving our figures. We agree that they will enhance our presentation and will implement these changes in the final paper.
>
> > In terms of pose estimation, I often hear about DeepLabCut…
>
> Including DeepLabCut, there are now numerous 2D and 3D animal pose estimation tools available which have enabled a wide variety of neuroscience studies (see L31-33 of our paper). We use these pose tracks as the features which are input into our (SC-)VAE models. So the contribution of our work is downstream of DeepLabCut-style analysis (i.e., it interprets the output of tools like DeepLabCut).
>
> In our contributed dataset here, we chose to use a pose estimation tool called DANNCE. For a more in-depth discussion of the differences, please see the DANNCE paper (Dunn et al. 2021).
>
> We will incorporate these suggested discussions. We thank the reviewer for helping us improve our paper.

---

> > ### Comment · Reviewer_p9qK · 2024-11-25
> >
> > I thank the authors for their response. I think the authors addressed all of the points that I raised and they agreed to incorporate the suggestions to their final paper. Thus, I raised my score to reflect these changes.

---

> > > ### Author Response · Authors · 2024-11-27
> > > **Following up with a revised manuscript**
> > >
> > > We thank the reviewer for their engagement. We have uploaded a revised PDF in which we have incorporated the suggested figure changes. We have also added some discussion in L534-538 of the conclusion on scrubbing in unknown datasets. We believe the paper is much improved thanks to the reviewer’s suggestions. We remain available for questions for the rest of the discussion period. And we will be happy to incorporate further discussions into the camera-ready version.

---

### Official Review · Reviewer_nAGW · 2024-11-04

**Soundness:** 4
**Presentation:** 4
**Contribution:** 4
**Rating:** 8
**Confidence:** 3

**Summary:**

The authors present a framework for motion analysis that uses conditional variational autoencoders to disentangle desired behavioral variables seen in animals and to remove nuisance confounds. They augment the C-VAE loss function to reduce dependence between the VAE latent variables and the behavioral variables of interest. They explore different methods to scrub out disentanglement including linear, quadratic, cubic, MLP, and MI based approaches. They thoroughly analyze their model in a simulated setting and apply their technique to improve disease detection in a Parkinsonian mouse model.

**Strengths:**

The paper is very clearly written, the results are compelling and thorough, and the model addresses a need in the neuroscience community as there are increasingly common adaptations of VAE style models to behavioral data. I appreciate the overview of C-VAEs as well as their motivation and clear explanation of their scrubbing methodology. The analysis on the two real datasets demonstrate the utility of the approach and I appreciate the clear visualizations demonstration appropriate disentanglement using the various SCVAES.

**Weaknesses:**

Some more discussion of the parkinsonian dataset would be appreciated. It is unclear what details are similar with this particular dataset and the previous one (e.g. are the behavioral variables scrubbed/conditioned the same way? are the model architectures and hyperparameters the same? ) This is a small point but it might be nice to point to supplemental information in this brief final results paragraph.

**Questions:**

Why do the authors think that the linear scrubbing improved the conditioned sequence best? This is seemingly the most limited way to represent z in equation 2? In principle, shouldn't the more sophisticated approaches (like quadratic and cubic) also be able to capture the linear approach? A bit more discussion on this point might be interesting to include.

---

> ### Author Response · Authors · 2024-11-18
> **Initial author rebuttal**
>
> We thank the reviewer for their time and thoughtfulness. We are encouraged to see that the reviewer appreciated the impact and timeliness of our work for neuroscience and found our results to be thorough and compelling in this application.
>
> > Some more discussion of the Parkinsonian dataset would be appreciated….
>
> We thank the reviewer for this suggestion. In case this was missed, some details are already included in Appendix B and C. However, we will expand these sections to include more details and clarifications about our datasets, model architectures, and hyperparameters. We will also include references to these sections in the main text.
>
> For the PD results in Fig. 4, we used a larger dataset (36 animals) in which animals were recorded before and after PD induction (via 6-OHDA injection). Otherwise, the data were collected in the same manner as the other figures. For the results in Fig. 4, we implemented animal identity scrubbing as described in Sec 4.2 “Animal Identity.” We reused the model architectures and most of the hyperparameters from the previous sections. The exception was the weight $\lambda$ on the scrubbing loss, which was re-tuned on the larger dataset (i.e., to minimize the decodability of $\mathbf{v}$ from $\mathbf{z}$ as in Fig. 2A-C).
>
> > Why do the authors think that the linear scrubbing improved the conditioned sequence best …
>
> This is a great question. Assuming that the reviewer is referencing the results in Table 1, we were not trying to suggest that linear scrubbers are definitively the best at this task. Note that the models used to produce the results in Table 1 and Fig. 3 had their hyperparameters tuned based on performance in Fig. 2A-C. It is possible that if tuned specifically for the motion synthesis task, the order of performance would change slightly.
>
> Our intent was to assess the difference between models without adversarial disentanglement (C-VAE), models using adversarial neural networks (SC-VAE-GR), and models which do not require adversarial neural networks (SC-VAE-MALS, -MAQS, and -MI). For our purpose, the larger-scale differences between these three categories of approaches were more important than the relative ordering of -MALS, -MAQS, and -MI. We show that even the simplest linear scrubber (-MALS) performs favorably to the neural network scrubber (-GR) with the additional advantage of not requiring the specification of a second neural network and its hyperparameters. We will update our writing in Sec 4.3 to clarify these points.
>
> If any additional questions or concerns arise, we will be happy to address them during this discussion period. In our revised paper, we will implement these changes and those suggested by other reviewers. Again, we thank the reviewer for helping us improve our paper.

---

> > ### Author Response · Authors · 2024-11-27
> > **Following up with a revised manuscript**
> >
> > As described in our initial response to this reviewer and in our most recent global response, we have expanded our discussion of scrubbing and improved our communication and analysis of the Parkinson’s dataset. We have also added more technical details on the datasets, architectures, and hyperparameters.
> >
> > We thank the reviewer again for their enthusiasm and constructive feedback. We remain available for the remaining discussion period to address further questions.

---

### Author Response · Authors · 2024-11-27
**Summary of Revised Manuscript**

We thank the reviewers for their time and insights. We are encouraged to see that all reviewers found our methods well-motivated and essential for our application in behavioral neuroscience. In response, we have uploaded an updated paper incorporating all the suggestions in the reviews. Our revision includes requested results from additional experiments and performance metrics, improvements to figures, edits to the text, and an expanded Appendix. We believe that our paper is much improved, and we thank the reviewers for this.

Here we highlight key improvements. A full itemized list of changes can be found on pages 27-28 of the revised PDF.

> New baseline - scrubbing with adversarial examples (SC-VAE-ND).

Reviewers qJjc and ZpeQ asked for additional baselines, specifically adversarial methods. Please note that there was a miscommunication in the initial paper, where we did not clearly convey that SC-VAE-GR was already included as an adversarial baseline. This has been clarified in the text.

However, we have also implemented an additional adversarial disentanglement method based on (Sanchez et al. 2020 “Learning disentangled representations via mutual information estimation.”) and have titled it SC-VAE-ND in the revised PDF. On all metrics except heading-conditioned motion synthesis, SC-VAE-ND performed worse than the previous adversarial SC-VAE-GR baseline.

We would like to re-emphasize that one of our primary contributions is to introduce scrubbers that: (1) do not require training a secondary network, and (2) can vary in assumptions on the level of scrubbing desired. These are our SC-VAE-MALS, -MAQS, -QD, and -MI scrubbers. To help clarify these contributions, we have updated some language at the end of our Introduction section, in Sec 4.4, and have reformatted our methods section to have a dedicated section (Sec 3.5) to discuss neural network scrubbers (SC-VAE-GR and -ND) and their associated challenges.

> Improving the presentation of figures and tables

We thank the reviewers for their suggestions on improving the presentation of our results. We have increased the size of all fonts and figure elements and polished the figure stylings. We have moved our former Figs. 1c and 2e to the Appendix and reformatted the former Fig. 4 into a table. Fig. 2a-c, Fig. 3b, Fig. 3f, and Table 1 have also been updated with results from the new SC-VAE-ND model. Table 2 (former Fig. 4) has been updated to better communicate how metrics relate to PD discriminability, and we have added new disease classification results. We have addressed all requested changes, and we are happy to make further changes for the camera-ready version if anything new is requested during the discussion period.

> Expanded Appendices

As suggested by reviewers nAGW and ZpEQ, we have expanded our Appendices to include more details on our calculated metrics, methods, model architectures, training, hyperparameter tuning, and datasets.

> A new control scrubbing model, new healthy/disease decoding metrics, and clearer writing for the Parkinson’s disease results.

In response to excellent suggestions from qJjc, we have made major improvements to the PD analysis section. In addition to rewriting the section in the paper to better motivate and interpret the analysis, we also added a new control model (scrubbing disease state) and a new healthy/disease classification accuracy metric (Table 2). We also distilled the MMD differences into a single effect size measure and simplified our partial lesion terminology (Table 2). We also further expanded on MMD and its definition in the Appendix. The Appendix also includes a new figure (Fig. 9) visualizing the MMD across animals and conditions for all models. We think this visualization elevates intuition for how MMD is changing when scrubbing animal ID and why this would increase healthy vs. disease discriminability.

> Clarifications to text and expanded discussion

Edits for clarity were made throughout (see pages 27-28 of the revised PDF for a detailed list). These edits included changes to better qualify statements made in the introduction and better explain the utility and novelty of parametric scrubbers and their different linear and nonlinear functional forms. We also expanded our discussion of limitations, addressing multivariate scrubbing and situations where nuisance variability may not be known.

Again, we sincerely thank the reviewers for helping us improve our paper. We believe we have addressed all reviewer concerns and look forward to any further discussion.

---

> ### Author Response · Authors · 2024-12-02
>
> We thank the reviewers once again for their helpful comments thus far. With the deadline approaching, please let us know if there are any unresolved concerns or questions that you would like us to address.

---

### Meta-Review · Area_Chair_9MYS · 2024-12-19

**Metareview:**

The paper proposes a framework for modeling 3D pose dynamics capable of disentangling both continuous and discrete nuisance factors, such as speed and animal identity. The authors employ a conditional variational autoencoder (VAE), where the conditioning variables correspond to the nuisance factors to be disentangled. To ensure proper disentanglement of these nuisance factors from the VAE latents, the authors introduce a series of strategies for "scrubbing" the nuisance factors from the latent representations. They demonstrate the utility of this approach in various downstream tasks, including clustering, decodability, and motion synthesis. Additionally, the technique is applied to enhance disease detection in a Parkinsonian mouse model, showcasing its broader applicability.

The paper is well-motivated, well-organized, well-written, and presents appealing results. The AC recognizes the contribution of the proposed framework and its promising results. After the rebuttal, all reviewers leaned toward accepting the paper. The AC concurs with the reviewers and recommends the paper for acceptance. To further enhance the quality of the paper, the AC encourages the authors to incorporate the reviewers' suggestions in the final revision.

**Additional Comments On Reviewer Discussion:**

Reviewer nAGW requested further discussion on the Parkinsonian dataset and an explanation of why linear scrubbing yielded the best results for conditioned sequences. Reviewer p9qK suggested improvements to the figure illustrations and raised questions about the generalizability of the method. Authors successfully addressed their concerns in the rebuttal.

Reviewer qJjc expressed concerns about the lack of comparisons with other established disentangling methods, unclear experimental details, and suggested rewriting the MMD analysis. Reviewer qJjc also posed some clarification questions. In the rebuttal, the authors provided a comprehensive response, including additional baselines, improved analysis of the Parkinsonian dataset, and enhancements to both text and figures. As a result, Reviewer qJjc increased the rating to 8.

In the initial review, Reviewer ZpEQ raised several concerns, including insufficient empirical or theoretical support, vague claims, inadequate baselines, unfair comparisons, and missing key implementation details. Following the first round of revisions, the authors addressed some of these concerns, leading the reviewer to increase the rating to 5 while highlighting remaining issues, such as the need for an ablation study to demonstrate the importance of each component within the scrubbers and their contribution to the overall model, as well as clarity in the revision. The authors further engaged in discussions with Reviewer ZpEQ and addressed these concerns. Ultimately, Reviewer ZpEQ was satisfied with the rebuttal and adjusted the score to 6, which is marginally above the acceptance threshold.

Overall, the authors have made significant revisions, addressing the reviewers' suggestions and concerns. All reviewers reached a consensus to accept the paper.

---

### Decision · Program_Chairs · 2025-01-22

Accept (Poster)